# Introducing Iterative Model Calibration (IMC) v1.0: A Generalizable Framework for Numerical Model Calibration with a CAESAR-Lisflood Case Study

Chayan Banerjee[1], Kien Nguyen*[1], Clinton Fookes[1], Gregory Hancock[2], and Thomas Coulthard[3]

[1]School of Electrical Engineering and Robotics, Queensland University of Technology, Australia
[2]School of Environmental and Life Sciences, University of Newcastle, Australia.
[3]Geography Department, University of Hull, UK.

**Correspondence:** Kien Nguyen* (k.nguyenthanh@qut.edu.au)

**Abstract.** In geosciences, including hydrology and geomorphology, the reliance on numerical models necessitates the precise calibration of their parameters to effectively translate information from observed to unobserved settings. Traditional calibration techniques, however, are marked by poor generalizability, demanding significant manual labor for data preparation and the calibration process itself. Moreover, the utility of machine learning-based and data-driven approaches is curtailed by the requirement for the numerical model to be differentiable for optimization purposes, which challenges their generalizability across different models. Furthermore, the potential of freely available geomorphological data remains underexploited in existing methodologies. In response to these challenges, we introduce a generalizable framework for calibrating numerical models, with a particular focus on geomorphological models, named Iterative Model Calibration (IMC). This approach efficiently identifies the optimal set of parameters for a given numerical model through a strategy based on a Gaussian neighborhood algorithm. Through experiments, we demonstrate the efficacy of IMC in calibrating the widely-used Landscape Evolution Model, CAESAR-Lisflood (CL). The IMC process substantially improves the agreement between CL predictions and observed data (in the context of gully catchment landscape evolution), surpassing both uncalibrated and manual approaches.

## 1 Introduction

Parameters of numerical (e.g. geomorphic) models play a crucial role in predicting their behavior. These models are usually calibrated based on observations at known data points or settings. However, it is often necessary to forecast how the system would behave at test data points or settings where direct observations are not possible.

A qualitative calibration approach involves a manual comparison of model and field data making it time-consuming and less likely to reveal the optimal model parameter configuration. On the other hand, a quantitative calibration of a numerical model involves assessing the model's error using statistics and is more suitable for complicated models with many parameters. Recently, there has been renewed interest in developing such automatic calibration routines to explore a model's parameter space (Becker et al., 2019; Brunetti et al., 2022; Beck et al., 2018; Tsai et al., 2021). Still, a large number of conventional approaches suffer from limitations like calibration of selective parameters, poor generalizability, extensive manual components

in data pre-processing and model calibration, and restrictive assumptions like differentiable and learning data-driven surrogate numerical models.

We propose a novel calibration algorithm: Iterative Model Calibration (IMC). The IMC is a fully automated calibration approach, which needs minimal manual interference and requires minimal data pre-processing. The method operates on a simple but effective concept of Gaussian-guided iterative parameter search. The process calibrates a defined list of parameters sequentially (high to low priority), with one parameter being adjusted at a time, keeping others fixed. The parameter values are sampled from a Gaussian neighborhood surrounding the latest parameter value. The model's output due to each predicted

parameter is then compared to the observed ground-truth data, and an error is calculated. This error serves as a fitness measure and a minimum threshold for finalizing the value corresponding to that particular parameter.

    In the following segment, we present a brief review of conventional approaches for calibrating geoscientific numerical models, specifically concerning LEMs such as CAESAR-Lisflood (CL). Some qualitative calibration strategies concentrate on one or a few chosen model parameters for calibration. For example, in (Ramirez et al., 2022), the focus was on the "m-

value" of CL's hydrology model (TOPMODEL), which is responsible for controlling the change in soil moisture storage for ungauged primary sub-catchments. They used a three-step approach: first, they ran a five-year simulation of the CL model with a 1km spatial resolution. Second, they repeated this process for a secondary sub-catchment, using the same rainfall input and calibrated parameters, lumped and spatially distributed. Lastly, they ran the calibrated primary sub-catchment hydrological model, which had spatially distributed m values, for a crucial short-term (3 h) extreme weather event, obtaining a simulated

discharge from the primary sub-catchment.

    In a study by Peleg et al. (2020), the hydrological TOPMODEL parameter "m" and Courant number were calibrated through selective calibration. This was done by finding an optimal fit between simulated hydrographs of 14 days and observed hydrographs. While carrying out this calibration, a number of parameters were manually set, with the help of published data from nearby locations and domain knowledge. In another work by Wang et al. (2022), CL calibration was carried out at selected

locations by reproducing the geomorphic changes and water depth driven by an extreme rainfall event. The parameter settings were set manually, based on domain knowledge and research data. Feeney et al. (2020) started with choosing CL parameter values from prior published literature. They then tested various combinations of the values to satisfy the two equations utilized in the lateral erosion algorithm in CL. Additionally, during calibration, they modified one parameter at a time while keeping the others constant. Skinner et al. (2018) employed the Morris Method on the CL model in two diverse catchments to discern the

impact of parameters on model behavior. Though centered on sensitivity analysis, this work indirectly aids model calibration by pinpointing key parameters for effective adjustments, thereby refining the calibration process.

    The tool described in (Beck et al., 2018) serves to calibrate the Lisflood hydrological model for designated catchment areas, deliberately omitting the upstream catchment region. It employs a genetic algorithm, LEAP, for the calibration process and is developed using Python. Nevertheless, a considerable amount of manual preprocessing of the input files, specifically scripts, is

necessary prior to initiating calibration runs. In contrast to previous approaches, Tsai et al. (2021) proposed a data-driven differentiable parameter learning (dPL) framework. This approach involves a parameter estimation module that maps raw input data to model parameters. These parameters are then fed into a differentiable model or its surrogate, such as a neural network-based

model. Differentiability allows for gradient calculation with respect to model variables or parameters, facilitating the discovery of hidden relationships in high-dimensional data through variable optimization. However, many physical or numerical models are not fully differentiable. Re-implementing a non-differentiable model into a differentiable one demands significant domain knowledge (Shen et al., 2023). Alternatively, a differentiable model can be developed from data using neural networks as surrogate models (Tang et al., 2020; McCabe et al., 2023), but this method requires extensive, often costly, field data collection and may struggle without specific historical data. These challenges limit the applicability and generalization of differentiable models and data-driven surrogates to complex numerical models like CL. A number of approaches leverage ML algorithms and general optimization algorithms for calibration. Brunetti et al. (2022) introduces a hybrid strategy calibration approach for hydrological models, combining precision ML algorithms like Marquardt–Levenberg with Comprehensive Learning Particle Swarm Optimization (CLPSO). Central to this approach is an objective function aimed at reducing the gap between HYDRUS model forecasts and empirical observations.

To sum up, the calibration of numerical models is hindered by reliance on extensive domain knowledge, manual tuning, and the high cost of data collection for ML approaches restricts their effectiveness and applicability. The expertise needed for model differentiation further limits widespread usage, underscoring the demand for adaptable and data-efficient calibration strategies in geoscientific modeling. A large number of conventional calibration techniques are tailored for hydrological models and have access to their wealth of data from global networks. But they fall short for geomorphological models (Abbaspour et al., 2004; Jetten et al., 2003) due to a lack of diverse and accessible data such as DEMs and information on soil, sediment, vegetation, and geology. This data scarcity undermines traditional calibration methods and hampers the use of newer data-driven ML in geomorphology, which depends on large datasets for accuracy. Our calibration approach aims to leverage limited DEM data to effectively calibrate geomorphological models, addressing a critical gap in current methodologies.

IMC algorithm introduces the following unique contributions:

1. Highly customizable approach: Due to the simplicity of the underlying process of iterative error-based search and parameter calibration, the algorithm is adaptable to any numerical model. Besides depending on the application, input-output files, and loss functions may be customized and substituted with ease.

2. Capable of calibrating a large number of parameters: The IMC is highly scalable and can calibrate for any number of numeric valued parameters of numerical models.

3. Minimal manual involvement requirement with a complete automated process: Apart from minimal data pre-processing and parameter initialization, IMC can run without any human supervision.

4. Generalizable for any numerical model: The algorithm doesn't have any restrictions regarding the type of numerical model. Being gradient-free, our approach requires neither the differentiability of the numerical model nor a neural network-based surrogate. With its generalization, it can be used as an add-on module and patched with any numerical model for calibration.

In the following sections we elaborate the IMC algorithm for calibrating numerical models, specifically targeting geomorphological models. We showcase the effectiveness of IMC by applying it to the Landscape Evolution Model, CAESAR-Lisflood, in the context of gully erosion modeling. The rest of the paper is structured as follows: Section 2 introduces the foundational concepts of model calibration techniques and establishes a general mathematical framework for addressing the problem. Section 3A is dedicated to a comprehensive exposition of our proposed IMC algorithm, including a detailed description of the algorithm itself and a discussion on each component of the IMC, referenced against the functional diagram shown in Fig. 2. In subsection B of Section 3, we offer a concise rationale for choosing Mean Square Error (MSE) as the metric for performance evaluation in our IMC algorithm. Section 4 outlines our case study, including the problem statement, details about the study location, and a discussion of the calibration results, supported by various tables and figures. In Section 5, we present a factual comparison of different calibration methods reviewed in this study against our IMC, complemented by an in-depth experimental analysis and additional experiments. The paper concludes with Section 6, where we summarize our findings and suggest promising directions for future enhancements to our work.

## 2    Preliminaries of Model Calibration

Calibration is an essential process in which the parameters of a model are adjusted to ensure that its output matches the observed historical data. The objective is to determine a set of parameter values enabling the model to produce data similar to the studied system (Oreskes et al., 1994; Gupta et al., 1998; Beven, 2006). Usually, a single fitness or loss value is sought to summarise the relationship between the predicted and observed data. As shown in Fig. 1, the model's parameters are adjusted repeatedly until the difference between the model output and the observed data is reduced below a certain threshold. Once a predetermined level of accuracy or error is attained, the calibration process is concluded and the model is deemed effective in simulating the real system or scenario.

When it comes to simple models, adjusting parameters and calculating errors is usually straightforward. However, numerical geomorphic models, e.g. Landscape Evolution Models, are more complex and have many configurable parameters. These model parameters can often have inter-related nonlinear effects on the model's behavior, making it challenging to anticipate how the model will behave with new parameter configurations (Skinner et al., 2018; Tucker and Hancock, 2010; Coulthard et al., 2007; Braun and Willett, 2013). As a result, doing trial and error matching of a model's parameters to specific field conditions is often complex, intricate, and time-consuming.

Furthermore LEMs often exhibit equifinality, where diverse parameter sets yield similar outcomes, highlighting the complexity of interpreting these models (Phillips, 2003). This phenomenon suggests multiple evolutionary pathways can lead to comparable landscapes, challenging model solution uniqueness and necessitating meticulous calibration and validation efforts (Beven and Freer, 2001). Additionally, equifinality may result in seemingly accurate landscape representations for incorrect reasons, pointing to the oversimplification of geomorphic processes (Lane et al., 1999).

Here we introduce mathematical notation to explain the calibration mechanism in general. Let $\mathbf{p}$ and $\theta$ denote the vectors of constant and calibration input parameters of dimension $d_1$ and $d_2$ respectively, of a certain numerical model $M$. Constant

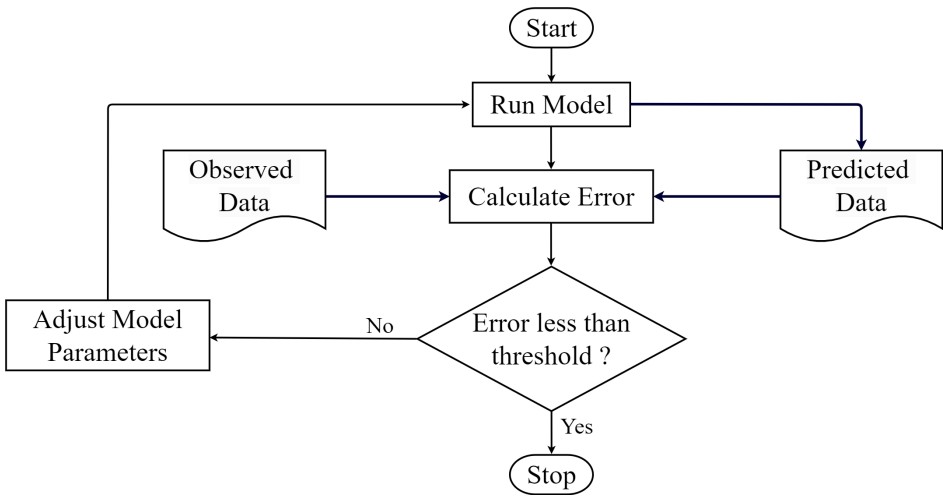

**Figure 1.** Overview of a typical calibration process.

input parameters stay the same over the whole calibration process, while the calibration parameters are sequentially optimized by the IMC algorithm. Also, let $S$ represent a collection of all input data, typically constituting DEMs, rainfall and soil data, etc. Formally we can describe the mapping of the constant, calibration input parameters, and input data to the expected model output as follows:

$$y(\mathbf{p},\theta,\mathbf{S}) = \eta(\mathbf{p},\theta,\mathbf{S}) + \xi$$

here $\xi$ represents the inherent randomness in the output of the numerical model, which is the uncertainty or variability that arises due to certain features within the simulation process. Sources of inherent randomness include system variability, incomplete knowledge, model imperfections, and numerical approximations. Here, the output of the numerical model is denoted by $y(.)$, which is a function of constant and calibration input parameters as well as input data. When calibrating a certain numerical model $(M)$, we assume we have certain information available to us.

1. The $n$ observations of the real system (e.g. natural processes) response $\mathbf{x} = \{x_1, \cdots, x_n\}$, corresponding to $n$ initial condition data $B_1 = \{\mathbf{S}_1, \cdots, \mathbf{S}_n\}$

2. The $n$ outputs generated by the numerical model $\mathbf{y} = \{y_1, \cdots, y_n\}$ for $n$ given input (initial condition) data and constant and calibration parameter vectors, i.e. $B_2 = \{(\mathbf{S}_1, \mathbf{p}_1, \theta_1), \cdots, (\mathbf{S}_n, \mathbf{p}_n, \theta_n)\}$.

The objective of the calibration algorithm is to iterative search for the unknown true calibration parameter vector $\theta^*$, which is the $\theta$ that parameterizes the numerical model to best match the observation of the real system or physical process. This naive calibration approach or direct calibration may be typically formulated as the following optimization problem:

$$\min_{\theta \in \Theta} L(x, y(\mathbf{p}, \theta, \mathbf{S}))$$

Where the goal is to find the $\theta$, such that it minimizes the above loss $L(.)$. The loss is calculated considering the observed response $x$ and the model generated output $y(\mathbf{p}, \theta, \mathbf{S})$.

## 3 Iterative Model Calibration (IMC)

### 3.1 Details of IMC algorithm

Fig. 2 presents a high-level overview of the interface of the IMC (proposed calibration algorithm) with the numerical model and their connection with other components and operations.

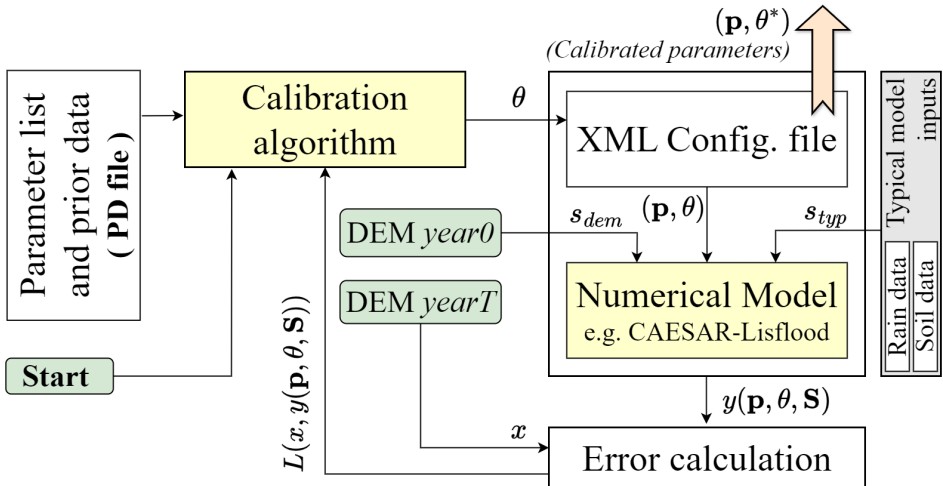

**Figure 2.** The calibration algorithm with a LEM numerical model presented here works in the following way. Firstly, the algorithm reads information regarding parameters from a PD file and updates the parameter suggestions in the XML file. Then, the numerical model reads the parameter values from the XML file and generates the output. The generated and target data are then compared, and the error is calculated based on a loss function. This loss is fed back to the algorithm, which uses it to set or update its loss threshold. The algorithm uploads a new parameter suggestion in the XML. This cycle continues until a stopping criterion is reached.

The following list briefly introduces the primary components of the setup:

1. Parameter list and prior data (**PD file**): contains a list of all the parameters $\theta$, that need to be calibrated. Along with the list, the file also contains prior best-known values of these parameters, the value's lower-upper limits $(up_\theta, lw_\theta)$ and

standard deviation $\sigma_\theta$ values, which is used to range the Gaussian search neighborhood. For more details on the contents and structure of this file, refer to Appendix and Table A2.

2. Calibration algorithm: is the proposed IMC algorithm that initiates by reading the calibrated parameter list, corresponding prior best-known values, value limits, and constraints (from the PD file) and outputs a new parameter value. This

parameter value is then passed on to the XML configuration file, which updates its parameter vector and forwards it to
the numerical model. The IMC later reads the error calculated from comparing the model output and observed data.

3. XML configuration file: holds the intermediate parameter values after being generated by the calibration algorithm. The numerical model reads the updated parameter vector from this file and generates simulated results accordingly. After the completion of the calibration process, the same file serves as the output, since the final calibrated values of the parameters are updated to the file.

4. Numerical model (M): is the model whose parameters are being calibrated. The model loads the constant and calibrated parameter vector sets $(\mathbf{p}, \theta)$ along with input data $S(= s_{dem} \cup s_{typ})$ and generates output $y(\mathbf{p}, \theta, \mathbf{S})$. Here $s_{dem}$ refers to the initial year DEM (i.e. DEM year-0) and $s_{typ}$ represents all the other types of typical data inputs that are loaded by the model e.g. Rainfall data and soil data.

5. Error calculation: Based on a predetermined loss function, this module compares the observed system response $(x)$ with
the model's output and quantifies the difference or similarity between them through a numerical value or score. We have used Mean Squared Error (MSE) as the error-generating function which is represented as follows:

$$L(x, y(\mathbf{p}, \theta, \mathbf{S})) = \frac{1}{a_1 \, a_2} [\Sigma_{q=1}^{a_1} \Sigma_{r=1}^{a_2} (K(q,r), P(q,r)]^2$$

where $x = K(.)$ and $y(.) = P(.)$ are ground truth and model predicted 2D numeric arrays respectively of dimension $i \times j$.

In the below explanation and the algorithm that follows, we have relaxed the dependence of $y$ on input data $\mathbf{S}$ from the notations but it is understood that outputs are with respect to these inputs.

In the IMC algorithm, each model parameter is numerically adjusted through a search process within its latest Gaussian neighborhood. A Gaussian neighborhood refers to the local region around a current parameter value, defined by the spread of the Gaussian distribution (typically within one standard deviation of the mean). Initially, the mean and standard deviation are
set as prior values, establishing a Gaussian distribution for each parameter. This distribution guides the exploration of parameter space during the calibration process. For each parameter $\theta_i \in \theta$ where $i = 1, \cdots, d_2$, the algorithm conducts a series of searches to find the optimal parameter value. Specifically, it performs $J \times C$ rounds of searching, where $J$ is the number of iterations for each parameter and $C$ is the number of rounds in each iteration.

The optimal parameter search is represented by *rounds*, where a model parameter value from it's latest Gaussian neighborhood
is selected and tested in the numerical model. Here, the parameter refers to the specific value being tested to see how well it performs. An *iteration* consists of a set of such rounds $(= C)$, representing multiple parameter searches. At the end of each iteration, if a better numerical model parameter is found that reduces the loss (beyond a certain threshold), the mean of it's Gaussian distribution is updated. This update process refines the distribution, improving the chances of selecting better numerical model parameters in future rounds. Therefore, the number of iterations represents the number of instances (for each
parameter) where the Gaussian distribution's parameter is considered for an update.

The calibration is sequential and while calibrating for a certain parameter say $\theta_i$ all other $\theta\backslash\theta_i$ parameter values are kept constant. Each $j^{th}$ iteration (where $j = 1, \cdots, J$) runs multiple rounds of random search in the Gaussian neighborhood of the last best-known parameter value. The Gaussian neighborhood is determined by the parameter's best-known value $\theta_i^b$ (known as prior information or passed on from previous iteration) and its fixed standard deviation $\sigma_{\theta_i}$, i.e. $\mathcal{N}(\theta_i^b, \sigma_{\theta_i})$. A randomly sampled data point ($\gamma$) from this neighborhood serves as the parameter value for the current round. It is also ensured that the sampled value $\gamma$ is well within the upper and lower value limits of the current parameter i.e. $up_{\theta_i} < \gamma < lw_{\theta_i}$.

Each iteration also keeps track of the best parameter value $\theta_i^{b',c}$ across its $C$ rounds, based on the minimum loss scored $L_{min}^c$. Besides a minimum loss threshold $\mathcal{L}$ is also maintained across all iterations and parameters. After each iteration if the $L_{min}^c < \mathcal{L}$ then its corresponding best parameter $\theta_i^{b',c}$ is saved as best value of the current parameter $\theta_i$ i.e. $\theta_i^b \leftarrow \theta_i^{b',c}$ and the min loss threshold is updated i.e. $\mathcal{L} \leftarrow L_{min}^c$. The whole process is elaborated as an algorithm as follows:

---

**Algorithm 1** The complete IMC algorithm

---

**Require:** Read parameter list $\theta$, their corresponding values (prior), s.d. ($\sigma_\theta$) and value limits $up_\theta, lw_\theta$ from file.

**Ensure:** Updated values for $\theta$ based on optimization criteria.

1: **for all** $\theta_i \in \{\theta\}$ **do**
2:    **for** $j = 1$ to $J$ **do**
3:       **for** $c = 1$ to $C$ **do**
4:          Obtain $\theta_i^c \leftarrow \gamma$, where $\gamma \sim \mathcal{N}(\theta_i^b, \sigma_{\theta_i})$ s.t. $lw_{\theta_i} < \gamma < up_{\theta_i}$
5:          Calculate $y(\mathbf{p}, \theta)$ where $\theta = (\theta\backslash\theta_i) \cup \theta_i^c$
6:          Evaluate loss $L^c = L(y(\mathbf{p}, \theta_i^c), x_i)$
7:          Update $\theta_i^{c+1} \leftarrow \theta_i^c$
8:          Save $(L_{min}^c, \theta_i^{b',c})$
9:          **if** $L_{min}^c < \mathcal{L}$ **then**
10:             Update $\theta_i^b \leftarrow \theta_i^{b',c}$ and $\mathcal{L} \leftarrow L_{min}^c$
11:          **end if**
12:       **end for**
13:    **end for**
14: **end for**

---

## 3.2   Choosing LEM performance evaluation metric

Assessing model performance is crucial for accurately depicting geomorphic changes. Choosing the right evaluation metrics, like the MSE of DEMs, is an efficient metric since directly measures topographic accuracy, a fundamental aspect of landscape studies.

LEM performance can be evaluated through various lenses, including erosion and deposition rates, sediment yield, hydrological accuracy, and more. These metrics serve to assess different facets of landscape dynamics and processes simulated by the model (Coulthard et al., 2002; Hancock and Willgoose, 2001; Tucker and Slingerland, 1997; Skinner et al., 2018; Barnhart et al., 2020; Skinner and Coulthard, 2023). Each metric focuses on specific attributes of landscape evolution, from the quantification of sediment transport to the replication of hydrological responses under varying climatic conditions. Notably, topographic accuracy emerges as a fundamental criterion, as it encapsulates the geomorphological fidelity of model simulations in replicating real-world landscapes (Temme and Schoorl, 2009).

The rationale for employing MSE between observed and LEM-estimated DEMs as a metric lies in its direct quantification of the discrepancy in topographical features. This approach allows for a granular assessment of model performance in simulating the spatial configuration of landscapes. Given the critical role of topography in governing hydrological and geomorphic processes, the accuracy of DEM simulations directly influences the reliability of LEM outputs in representing erosion patterns, sediment transport, and hydrological dynamics.

Moreover, the use of MSE aligns with the principle of evaluating model efficiency through quantitative measures that provide clear benchmarks for improvement (Nash and Sutcliffe, 1970). By quantifying errors in elevation across the landscape, MSE offers a comprehensive overview of model performance in capturing the intricate details of terrain morphology.

Additionally, the comparison of DEMs through MSE facilitates the identification of systematic biases or inaccuracies in model simulations, guiding further calibration and refinement of LEM parameters (Beven and Binley, 1992). This aspect is particularly crucial in landscape evolution modeling, where the spatial distribution of elevation changes significantly influences erosion and sedimentation processes.

## 4 Case Study: Calibration of LEMs for predicting Gully Evolution

### 4.1 Problem statement

Our primary objective is to calibrate the numerical model (here CL) using geomorphological data from two distinct years, 2019 and 2021, including DEMs, soil, and rainfall data. IMC calibration aims to enhance the model's reliability by ensuring its outputs closely match observed data. Achieving this alignment is essential for gaining accurate insights into landscape evolution dynamics.

Additional objectives include comparing our calibration method with existing approaches to highlight its broader applicability and reduced human effort. We aim to conduct experiments with varying calibration run lengths to assess their impact on calibration quality, focusing on erosion volume and spatial accuracy. Furthermore, we seek to evaluate the efficiency of the proposed IMC algorithm in re-estimating known parameter values from deliberate perturbations, demonstrating its accuracy and robustness.

## 4.2 Study area and data

The study area is a gully catchment region situated 20 km to the east of Mount Abbot National Park (Scientific) in the Bowen Basin region of Northern Queensland, at a location: $20°13'S, 147°33'20''E$, see Fig. 3. For hourly rainfall data (see Fig. 3(b)) we have used pluviometer reading from Ernest Creek Pluvio of Burdekin basin, Queensland (WMIP, 2024), between the dates $1^{st}$ July $2019 - 2021$. The DEMs are collected using Airborne Laser Scanning (ALS) by the Department of Agriculture Water and the Environment, Australia under project names Bogie 2019 and Strath Bogie 2021 and hosted on an online repository (ELVIS, 2024). The required DEMs are downloaded from the mentioned source with the following specifications: Resolution: 0.5m, Vertical Accuracy: $\pm 0.15\,\mathrm{m}\,@\,67\%\,\mathrm{CI}$, Horizontal Accuracy: $\pm 0.3\,\mathrm{m}\,@\,67\%\,\mathrm{CI}$. For ease of computation, we have used a downsampled version (i.e. 1m) of the original DEMs, in all our experiments.

We chose gully erosion in Australia as a case study due to its environmental significance, the availability of extensive data, and the unique challenges posed by Australia's climate and soil. The study aims to inform local policymakers and land managers, fill research gaps, and develop targeted strategies for erosion mitigation. Additionally, the insights gained from this specific context can illustrate the framework's adaptability and transferability to other regions facing similar environmental challenges.

## 4.3 Calibration experiments and results

In the following sections, we introduce the study area and present the essential parameters and settings used for running IMC in CL parameter calibration. Additionally, we provide comparative results from the experiments, including CL with uncalibrated parameters, CL with manually calibrated parameters, and CL with Manual + IMC calibrated parameters.

### 4.3.1 Calibration details and experimental setup

We present Table 1, which summarizes essential information regarding the primary parameters of the CL numerical model, including numerical values from existing literature. Additionally, the table shows the prior values used to initialize the IMC for each parameter to be calibrated in the *PD file*. In the IMC's calibration process, the loss function is very important. As mentioned in Section 4, we consider the MSE of ground truth target data and CL predicted data in image format, for calculation of error at each round. We explore different forms of ground truth and CL-predicted data (such as DEMs and DEM of difference i.e. DoD) and show how they can be purposed for specific experimentation.

Our primary experiments investigate the effectiveness of the IMC approach in calibrating the parameters of CL, with a particular focus on accurately predicting erosion volume. This is important because erosion volume impacts landform stability, environmental health, and cost-effectiveness, and is significant for landform design and risk assessment. We use the input and predicted DEMs (i.e. $DEMyear0$, $DEMyearT$ and $D\hat{E}MyearT$), to generate the target and predicted difference of DEMs

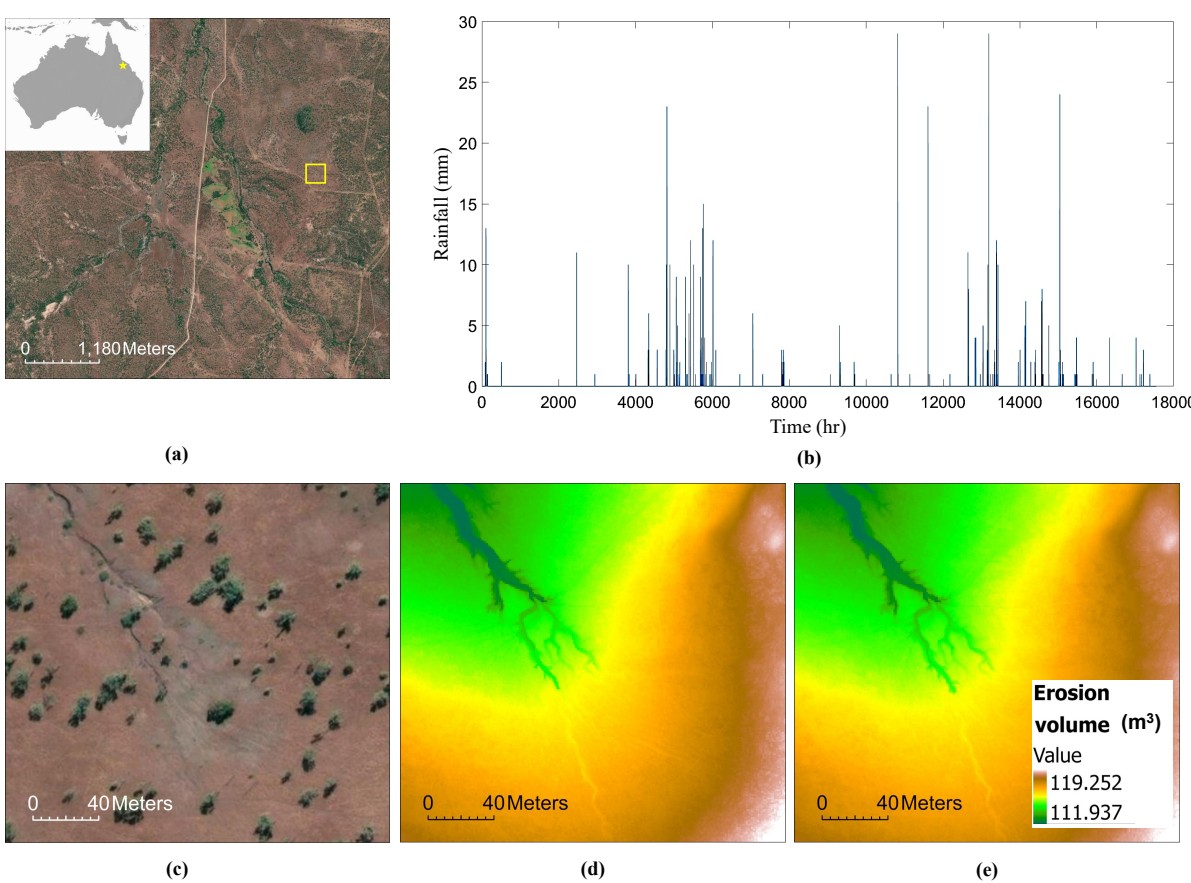

**Figure 3.** Study site information (a) Satellite image of the Bowen Basin, with the yellow box highlighting the study location. Inset shows the location of the Bowen basin (yellow star) in Australia. (b) Hourly rainfall data between July 2019-2021, pluviometer reading. (*Source: Basemap and data provided by Esri and its Community Map contributors. Pluviometer reading from Ernest Creek Pluvio of Burdekin basin, Queensland (WMIP, 2024), between the dates $1^{st}$ July* $2019-2021$). (c) Magnified (zoomed-in) view of the study region (d) Observed DEM of year 2019 in colormap, it is used as CL's input. (e) Observed 2021 DEM in colormap.

**Table 1.** Primary CL parameters, their values from manual, literature, and their model's sensitivity. Sensitivity scoring uses asterisks (*) to indicate the impact of parameters on model outcomes, from very high "***" to low "*". The table also presents the default parameter and prior values assumed by the IMC algorithm.

| Parameter Names | Model Sensitivity (Skinner et al., 2018) | CL Reference Manual | Tin Creek (Australia) | IMC algorithm Search Range | IMC priors $(\mu, \sigma)$ |
|---|---|---|---|---|---|
| Max erode limit (m) | *** | $0.01(10m)$ | $0.001 - 0.003$ | $0.001 - 0.01$ | $0.003, 0.001$ |
| In-channel lat. Erosion | *** | $5 - 50, 200 - 1000$ | $10.0 - 30.0$ | $10.0 - 30.0$ | $20, 05$ |
| Vegetation crit. Shear stress (Pa) | *** | Not specified | $2.0 - 7.0$ | $2.0 - 7.0$ | $3, 1$ |
| Min Q for depth calculation (m) | *** | DEM resol./ 100 | $0.025 - 0.075$ | $0.009 - 0.01$ | $0.01, 0.001$ |
| Slope failure threshold ($^\circ$) | *** | Not specified | $40 - 50$ | $40 - 60$ | $50, 5$ |
| Evaporation rate (m/day) | *** | Not specified | $0.0025 - 0.01$ | $0.002 - 0.01$ | $0.005, 0.001$ |
| Soil creep rate (m/yr) | ** | $0.0025$ | $0.00125 - 0.00375$ | $0.001 - 0.004$ | $0.0025, 0.001$ |
| In-out difference allowed (m$^3$/s) | ** | Not specified | $0.1 - 0.4$ | $0.1 - 0.4$ | $0.2, 0.1$ |
| Slope for edge cells | ** | Not specified | $0.0025 - 0.0075$ | $0.002 - 0.01$ | $0.005, 0.001$ |
| Manning n | ** | Variable | $0.03 - 0.04$ | $0.005 - 0.2$ | $0.01, 0.001$ |
| Grass maturity rate (yr) | * | from 0 to 1 | 0.5 - 2.0 | $0.1 - 2.0$ | $0.5, 0.1$ |
| m value | * | $0.02, 0.005$ | - | $0.005 - 0.02$ | $0.01, 0.001$ |

i.e. $DoD$ and $\hat{DoD}$ as follows:

$$DoD_{Target} = DEMyear0 - DEMyearT$$
$$DoD_{Predicted} = DEMyear0 - D\hat{E}MyearT$$

In order to focus the calibration on the erosion volume we multiplied the DoDs with a mask ($= m(e, f)$), which can be defined as follows:

$m(e, f) = 0, val(e, f) < 0$
$$= 1, val(e, f) > 0$$

where $(e, f)$ represents a location on a DoD and $val(e, f)$ represents the signed magnitude of that data-point. Such that the final DoDs can be written as

$$DoD_{Target} = DoD_{Target} * m(e, f),$$
$DoD_{Predicted} = DoD_{Predicted} * m(e, f).$

In later experiments (Section 5.3) we also investigated the accuracy of IMC-based calibration of CL's default parameters. In the experiments we try to estimate a single parameter at a time from a perturbed value, keeping all other parameters fixed. In that context we have simply considered the following:

$$DEM_{Target} = DEMyearT$$
$DEM_{Predicted} = D\hat{E}MyearT$

See the relevant section for more details on the experiments.

The code for the proposed IMC algorithm with CL as a case study, along with the data used in this study, is archived for easy accessibility (Banerjee, 2024).

### 4.3.2 Calibration results

In this section, we present and discuss the results of the calibration process. Comparative results are presented in Table 2 and Fig.4, highlighting the differences between the CL model results obtained using different variations of calibrated and uncalibrated parameter sets. For the *uncalibrated* set, we consider the default CL parameters and simply adapt them to our study area and DEM dimension. In the *(Manual calibration)* set, we use existing literature-based knowledge of parametric values w.r.t the study area and update the default CL parameter set. Finally, in *Manual + IMC* set (also referred to as IMC for

brevity), we start or initialize the IMC calibration process with the *(Manual calibration)* set data. Additionally, Table3 provides the comprehensive results of the IMC calibration process for all CL parameters, across three separate calibration runs of the same length ($5 \times 5$).

    In detail, Table 2 numerically shows that IMC-based calibration of CL parameters encourages the CL to predict future erosion volume with substantial accuracy as compared to the CL's results with uncalibrated and manually calibrated parameters. We

also show that using only basic knowledge of the value range of parameters of the study region, two temporally separated DEMs (i.e. 2019 and 2021), and the rainfall data over this period the IMC can calibrate the CL parameters, evident by its prediction of the target erosion volume. The target erosion volume is derived from the difference between the 2021 DEM and the 2019 DEM.

**Table 2.** Comparison of Total Erosion Volume and corresponding MSE loss: Observed data vs. CL Using Uncalibrated, Manually Calibrated, and IMC Calibrated Parameters. The results presented below are from three separate calibration runs, each with fixed-length runs ($5 \times 5$) and taking around 5 hrs. Due to the stochastic nature of the calibration process, the mean values are reported along with their standard deviations (mean ± std. dev.).

| Case | Erosion volume (m$^3$) | MSE |
|---|---|---|
| DoD Observed | **49.551** | – |
| DoD Uncalibrated parameters (CL config. adapted for 1m DEM, DEM ours, rain ours) | 21.756 | – |
| DoD Calibrated parameters (manual) | 38.819 | – |
| DoD Calibrated parameters (manual + IMC) [Mean± std] | $50.068 \pm 2.161$ | $0.000333 \pm 2.1 \times 10^{-6}$ |
| DoD Calibrated parameters (manual + IMC) [Best] | **51.495** | **0.000328** |

**Table 3.** CL parameters calibrated via IMC across three separate calibration runs of fixed length ($5 \times 5$), denoted as "Run01, Run02, and Run03". An "IMC initial value" column presents the parameter initialization value for each run. The last two columns display the mean with standard deviation, and the coefficient of variation (CV). The CV, a standardized measure of dispersion, is defined as the ratio of the standard deviation to the mean, expressed as a percentage. It is useful for comparing the relative variability of parameters with different units or scales. High variability is observed in parameters 1 to 4, indicated by higher CV values (see also Fig. 5(a)). The concluding row, showcases MSE loss, which identifies "Run03" as the optimal calibration run.

| Sl no. | Parameter names | IMC | Separate IMC calibration runs | | | Mean $\pm$ Std. deviation | Coefficient of Variation (%) |
|---|---|---|---|---|---|---|---|
| | | initial value | Run_01 | Run_02 | Run_03 | | |
| 1 | slope of edge cell (initialq) | 0.005 | 0.00452 | 0.00657 | 0.00238 | $0.0045 \pm 0.0020$ | 45.37 |
| 2 | Max erode limit (m) | 0.3 | 0.0087 | 0.005 | 0.00358 | $0.0057 \pm 0.0026$ | 45.61 |
| 3 | Evaporation rate (m/day) | 0.005 | 0.004 | 0.005 | 0.00905 | $0.0060 \pm 0.00267$ | 44.50 |
| 4 | In-out difference (initscans) (m³/s) | 0.2 | 0.2619 | 0.15967 | 0.11447 | $0.1787 \pm 0.0755$ | 42.24 |
| 5 | In-channel lat. Erosion | 20 | 15.554 | 12.893 | 24.1937 | $17.5469 \pm 5.9080$ | 33.67 |
| 6 | Grass maturity rate (yr) | 0.5 | 0.3509 | 0.515 | 0.6736 | $0.5131 \pm 0.1613$ | 31.43 |
| 7 | m value | 0.01 | 0.00687 | 0.0118 | 0.00862 | $0.0091 \pm 0.0024$ | 26.37 |
| 8 | Manning n | 0.01 | 0.0111 | 0.012 | 0.00714 | $0.0101 \pm 0.0025$ | 24.55 |
| 9 | Vegetation crit. Shear stress (Pa) | 3 | 4.0934 | 2.765 | 3.6401 | $3.4995 \pm 0.6752$ | 19.29 |
| 10 | Soil creep rate (m/yr) | 0.0025 | 0.00343 | 0.0039 | 0.00397 | $0.0038 \pm 0.0003$ | 8.47 |
| 11 | Min Q (m) | 0.01 | 0.00965 | 0.0097 | 0.01059 | $0.0099 \pm 0.00052$ | 5.25 |
| 12 | Slope failure threshold (°) | 50 | 41.2818 | 40.372 | 40.2236 | $40.6258 \pm 0.5729$ | 1.41 |
| | | | | | | | |
| - | **MSE loss** | - | 0.000332 | 0.000329 | **0.000328** | - | - |

In Fig.4, we further elaborate on the numerical results presented in Table2 through extensive visual comparison. Here, we compare the CL's prediction of erosion volume using three different sets of parameters: *uncalibrated*, *manual*, and *manual + IMC*. The results demonstrate that the combination of basic manual calibration with the automated IMC process significantly enhances CL's accuracy in predicting the target erosion volume.

In Fig. 5(b) we present a detailed side-by-side comparison of DoD value distributions across various calibration settings, benchmarking them against observed DoD values. To effectively summarize statistical variations in DoD, we use boxplots representing 1D vectors derived from flattened 2D DoD arrays. These boxplots offer a concise visual summary of central tendencies, spread, and outlier behavior across calibration settings, allowing us to assess how each calibration scenario aligns with observed DoD and identify systematic biases or deviations.

The comparison of DoD values across different calibration settings reveals improvements with the *Manual + IMC approach*, which better approximates *observed DoD* values. The *uncalibrated* model's DoD shows a narrow whisker range ($-0.002460, 0.0024$) and a median of $-0.000080$, reflecting minimal variability (IQR $= 0.001214$) and suggesting underestimation of observed terrain changes. In contrast, in the *manually calibrated* model the median shifts to $0.000024$ and broadens the IQR to $0.002129$, capturing more terrain variability.

The *Manual + IMC* approach further refines the model, with a near-zero median ($-0.000001$) and expanded IQR ($0.002826$), increasing sensitivity to subtle changes. While observed data display a much wider IQR ($0.056000$) and whisker range ($-0.270000, -0.0460$) reflecting significant erosion, the widened whiskers ($-0.005645, 0.005661$), in *Manual + IMC* ($133.44\%$ broader than *uncalibrated* and $33.02\%$ than *Manual* calibration approaches) shows improved robustness. Thus enabling reflec-

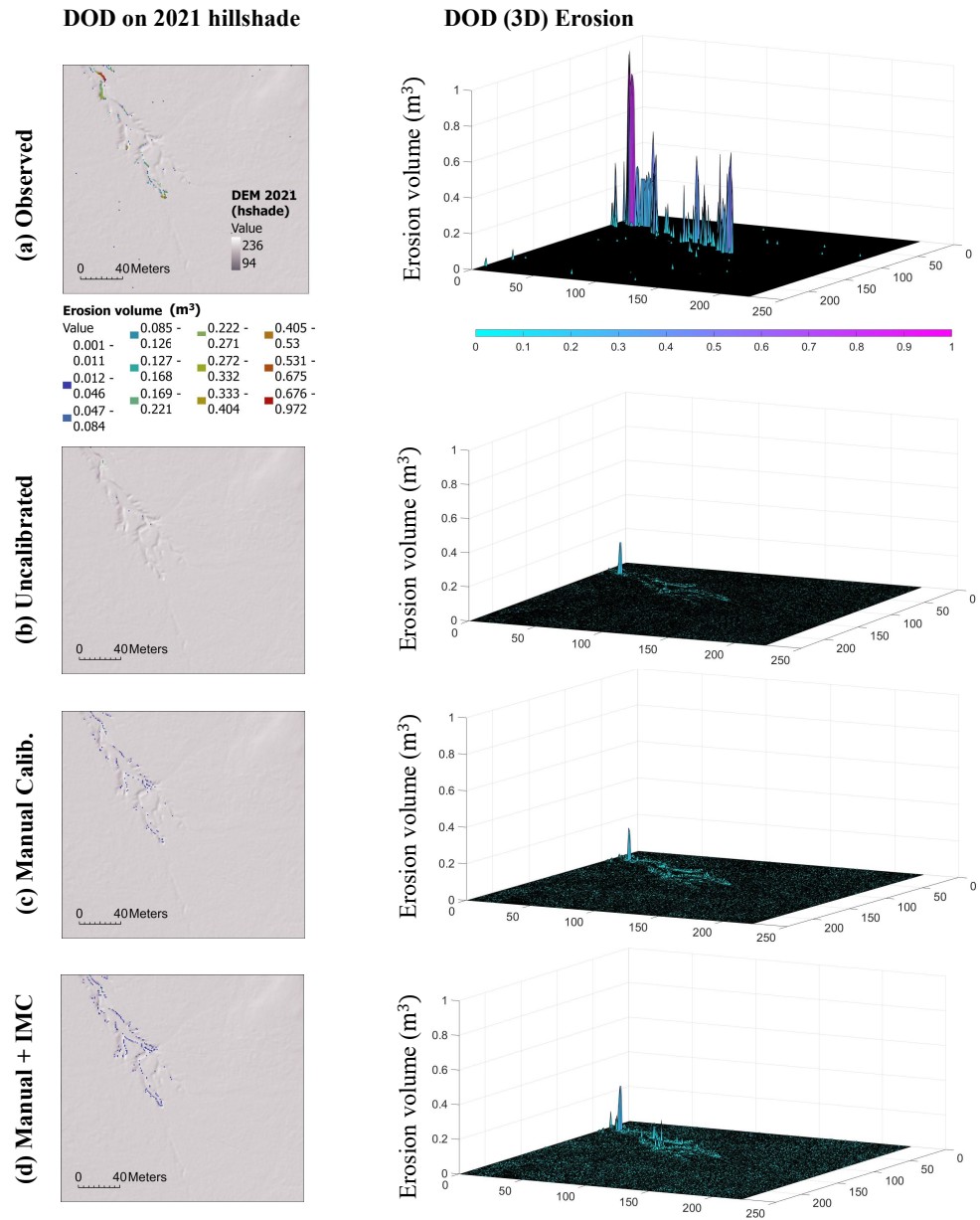

**Figure 4.** The left column shows the DoD between the 2019 and 2021 DEMs, focusing only on erosion volume. This erosion data is overlaid on the 2021 DEM hillshade to provide spatial context. Areas with nearly zero erosion volume are shown as transparent to highlight regions with more significant erosion. The right column presents corresponding 3D plots of the DoD, focusing on erosion volume. Compared to all other approaches, the *manual+IMC* calibration *(d)* show the closest resemblance of erosion volume to the *Observed (a)*, both spatially and volumetrically. (*DEM Source: (ELVIS, 2024)*).

tion of real-world geomorphic changes more accurately, improving the sensitivity to natural variations and localized changes in terrain.

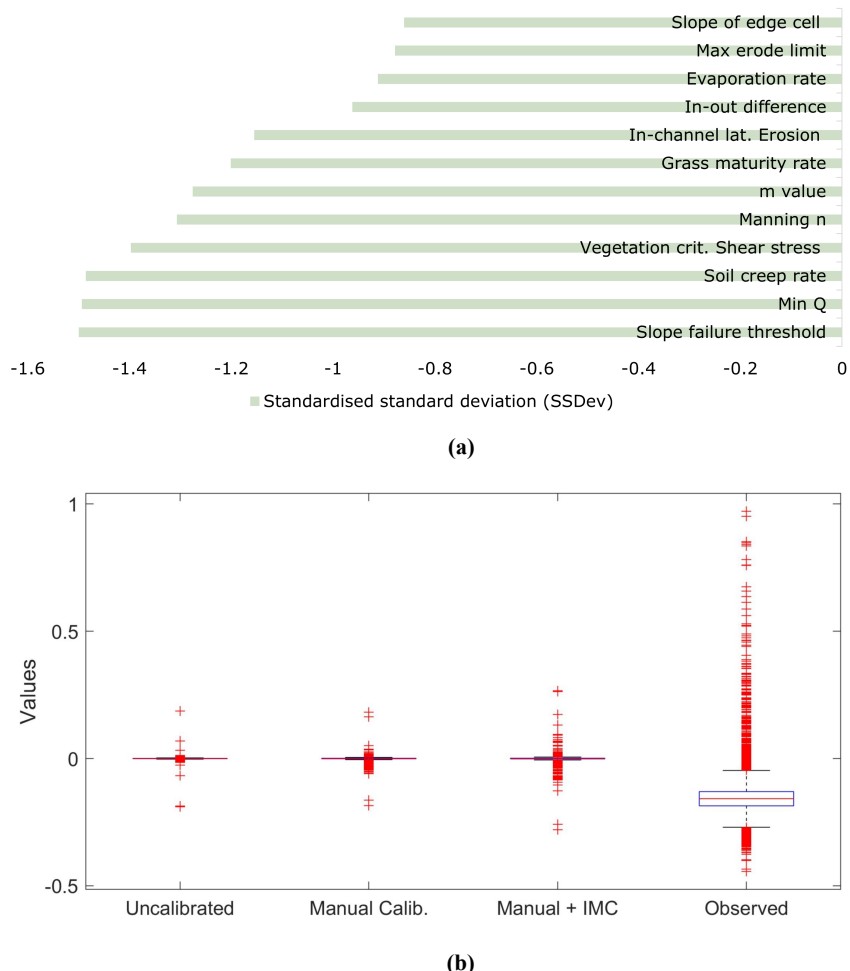

**Figure 5.** (a) Comparison of parameter variability across three calibration runs, using standardized standard deviation (SSDev). SSDev represents the z-score of the standard deviation of each parameter's repeated experiment (three individual calibration) values. This metric quantifies how many standard deviations a parameter's variability deviates from the mean variability of all parameters, facilitating a direct comparison of consistency and stability among different parameters. Lower SSDev values indicate parameters with variability below the average, signifying higher consistency, while higher SSDev values indicate greater variability relative to the repeated experiment average. (b) Comparison of DoD data distribution across various calibration settings of CL Parameters. The *uncalibrated CL* data clusters near zero, demonstrating low accuracy. In contrast, *manual* calibration—especially when combined with *IMC*—enhances the results, capturing greater variability in the data.

## 5 Comparisons and Experimental analysis

 ### 5.1 Comparison with existing calibration approaches

A majority of calibration approaches surveyed so far calibrate for specific and partial parameters only, involve a consider-able human effort towards parameter value selection/customization (Wang et al., 2022; Peleg et al., 2020; Ramirez et al., 2022; Feeney et al., 2020), and operate for a particular type of numerical model e.g. Lisflood (Beck et al., 2018), CAESAR-Lisflood(CL) (Wang et al., 2022; Peleg et al., 2020; Ramirez et al., 2022; Feeney et al., 2020), HYDRUS (Brunetti et al., 2022)
 and Victoria(Tsai et al., 2021) numerical models.

The usability and generalizability of a certain approach directly depends on the set of input data required during parameter calibration. The requirement of data in addition to the ones used by the target numerical model increases the complexity to adapt the calibration for different settings and adds a heavy overhead. The following table summarizes the differences between the existing calibration approaches for LEMs, specifically CL.

**Table 4.** Comparison of different calibration approaches

| - | Input files and preliminary assumptions | Param. calibrated | Manual comp. | Target Model |
|---|---|---|---|---|
| (Beck et al., 2018) | TS observed discharge, Static maps (DEM, land use, etc.) TS input meteo variables over calibration period | All | High | Lisflood |
| (Wang et al., 2022) | Typical CL inputs | Hydrology param. | Very high | CL |
| (Peleg et al., 2020) | Typical CL inputs, hydrograph | Hydrology param. | Very High | CL |
| (Ramirez et al., 2022) | Typical CL inputs | Hydrology param. | Very High | CL |
| (Feeney et al., 2020) | Typical CL inputs | Partial | Very high | CL |
| (Skinner et al., 2018) | Typical CL inputs | All | Low | CL |
| (Brunetti et al., 2022) | Hydrology parameters | - | Low | HYDRUS Simunek et al. (2016) |
| (Tsai et al., 2021) | Typical model inputs Differentiable model or NN-based model surrogate | All | Low | VIC model (Hamman et al., 2018) |
| Ours | Typical CL inputs | All | Very low | CL (customizable) |

 ### 5.2 Experimental analysis

In this section we discuss experiments with different lengths of calibration runs, which is equal to the total rounds ($= rounds \times iterations$) of calibration operated per parameter (see Fig. 6); refer to section: 3.1, for the explanation on the terms *round* and *iteration*. It is important to understand that the quality of calibration of CL parameters using IMC would be reflected through a couple of quantities. First, the proximity of the predicted and the observed DoDs in terms of the total volume of erosion
 (numerically). Second, both volumetric and spatial similarity of the erosion and their location of occurrence, are quantifiable by the MSE loss.

Also, the similarity of total erosion volume of the predicted and Observed DoDs/ DEMs doesn't alone guarantee actual similarity and they still may be far apart if their MSEs are substantially different. This phenomenon can be seen in Fig. 6b, where the calibrations with lesser calibration (i.e. 2x5 and 5x2) duration though have a close enough erosion volume to the

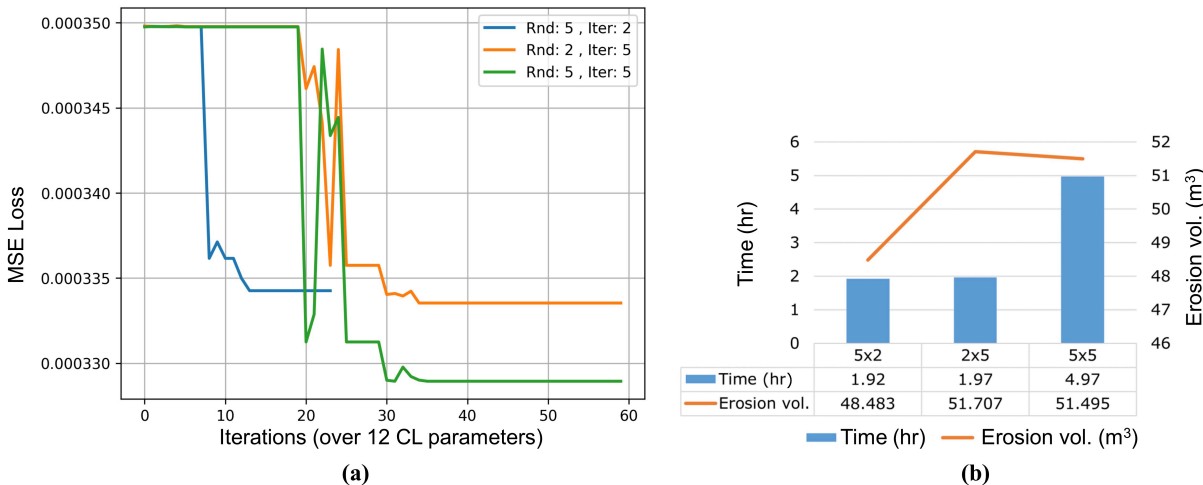

**Figure 6.** Calibration run/ duration (a) Shows comparison of error or calibration losses for different lengths of calibration runs ($= rounds \times iterations$) run for 1m resolution DEM (b) Shows a side-by-side comparison of the total time taken by different calibration runs and the erosion volume achieved (target being 49.551).

observed but show higher MSE. This portrays that the parameter exploration has been inadequate and due to the selection of sub-optimal parameters the end resulting erosion volume though numerically similar is spatially misplaced or distributed on the surface.

## 5.3 Further experiments: evaluating IMC's Efficiency in CL parameter re-estimation

In this experiment, we want to show how accurate and efficient IMC is at re-estimating known ( we refer as Benchmark)
parameter values for CL software after deliberately changing them. These known parameter values are the default settings provided with the CL software distribution. We use the default CL parameters, the initial DEM ( as $DEM_{year0}$ ) and other provided data and create a future ( or $DEM_{yearT}$) DEM. Next, we use these two DEMs to re-estimate the parameters with IMC, starting from their deliberately perturbed versions. we intend to show that IMC can accurately return to the known parameter values.
To ensure the experiment remains both insightful and manageable, we focus on two key parameters: Maximum erode limit and Lateral erosion rate. They are selected due to CL's pronounced sensitivity to these, as seen in (Skinner et al., 2018) and listed in Table 1. In this experiment, we start with producing a target DEM (i.e. $DEM_{yearT}$, where $T = 2$ years) entirely using CL's default parameter and dataset (provided with distribution (Coulthard et al., 2024)). Next, we individually alter each of these two key parameters mentioned earlier, maintaining the rest at their original values. Subsequently, we employed the IMC
algorithm to accurately estimate the true values of these parameters from their altered states.

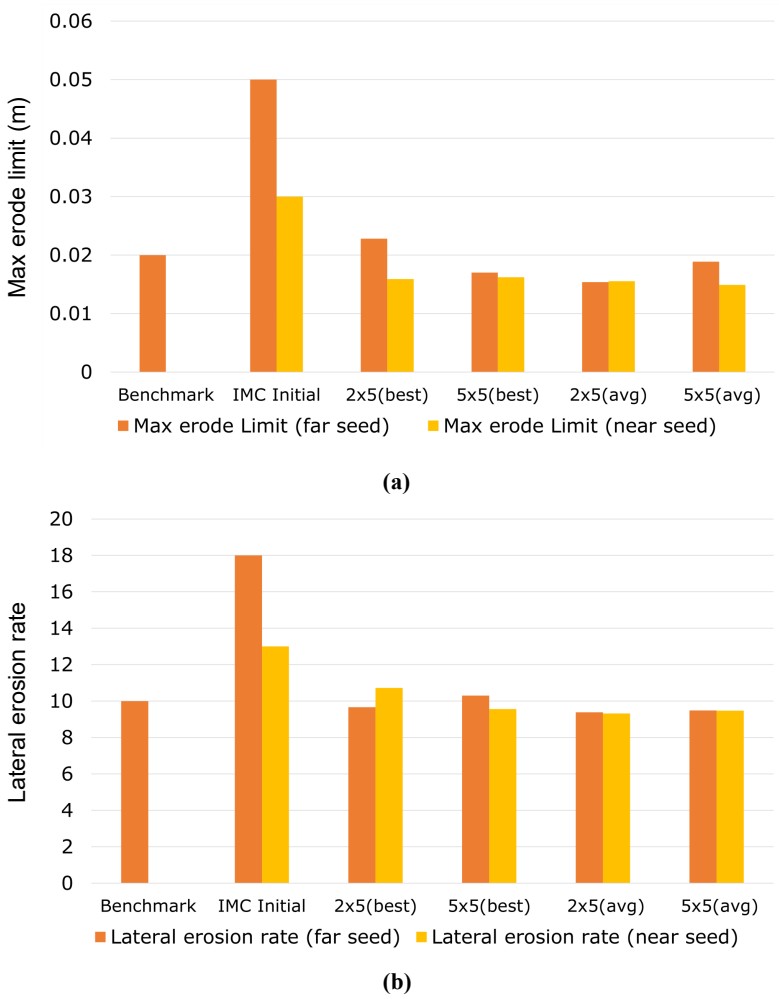

**(a)**

**(b)**

**Figure 7.** Estimating known numerical values (Benchmark) of CL parameters from their deliberately perturbed versions (a) Estimation of *Max erode limit* parameter (b) Estimation of *Lateral erosion rate* parameter. *Benchmark* refers to the "known" CL parameter value and *IMC initial* is the perturbed version of the same, from where the IMC starts calibrating. IMC is run at different lengths ($= round \times iteration$) repeatedly and the best and average (of three separate calibration runs)of estimated parameter values are presented.

**Table 5.** Calibration data regarding CL known parameter re-estimation experiment, detailed in Sec 5.3

|  | **Benchmark value** | **IMC initial** (far seed) | **IMC initial** (near seed) | **IMC search range** | **IMC priors** $(\mu, \sigma)$ |
|---|---|---|---|---|---|
| Max erode limit (m) | 0.02 | 0.05 | 0.03 | 0.01 - 0.06 | (IMC initial, 0.01) |
| Lateral erosion rate | 10 | 18 | 13 | 8 - 20 | (IMC initial, 5 ) |

The parameters are estimated through individual IMC calibration runs, which are repeated three times to account for the stochastic nature of the process. The mean value of the repeated runs is calculated and presented alongside the best value, which is closest to the observed. Please refer to Fig. 7 for a visual representation of this data.

At the beginning of each calibration, we set the values of the "Maximum erode limit" and "Lateral erosion rate" parameters to their respective "IMC initial" values, which are deliberately perturbed from observed values. We conducted the experiments using "IMC initial" values selected from positions both proximal (termed "near seed") and distal (termed "far seed") relative to the observed values of each parameter. This approach was designed to affirm IMC's effectiveness irrespective of the initial proximity of the "IMC initial" values to their observed counterparts.

The experimental outcomes are detailed in Fig. 7, with corresponding calibration data provided in Table 5. These results illustrate IMC's capability to accurately re-estimate the true values of both the parameters. Specifically, for the *Maximum Erode Limit*, we observe a minimum absolute error of $0.0028 (= |Benchmark - Estimated|)$, with the best-estimated value being $0.0228$ (2x5(best)) compared to the observed value of $0.02$. In the case of the *Lateral Erosion Rate*, the minimum absolute error recorded was $0.302$, where the best-estimated value reached $10.302$ (5x5(best)), closely aligning with the observed value of $10$.

The slight deviations in accurately estimating the observed parameter values can potentially be linked to the sensitivity of the MSE loss function to noise, wherein minor discrepancies could be amplified into seemingly larger differences. Moreover, the intricate nonlinear relationship between a parameter in the CL model and its resultant geomorphic output can occasionally lead IMC into local optima traps. These challenges could be mitigated by adopting a tailored loss function specifically designed to capture the complex geomorphological dynamics more effectively. Additionally, incorporating strategies such as stochastic perturbation and advanced optimization techniques may facilitate overcoming the hurdles of local minima, thereby enhancing the fidelity of parameter estimation in geomorphological simulations.

## 6  Conclusions

This study introduces a versatile, adaptable, and scalable calibration algorithm for numerical models, demonstrated through its application in calibrating the Landscape Evolution Model: CAESAR-Lisflood. The outcome of this calibration is the generation of geomorphic data for a gully catchment landscape evolution scenario, with significantly closer predictions to observed data, compared to uncalibrated and manual approaches.

The proposed calibration technique is adaptable to various numerical models and requires minimal extra input beyond conventional CL inputs. However, it has its limitations. Although erosion volumes are similar to target patterns in both space and volume, discrepancies remain. Specifically, the "*Manual + IMC*" approach tends to spread erosion volume across the study area in small amounts, affecting calibration precision. Additionally, the calibration process is inherently stochastic, resulting in non-unique, varying parametric vectors across calibration sessions, even under identical conditions. We used Mean Squared Error (MSE) for its ease and ability to emphasize large errors, widely applied in areas such as computer vision. However,

MSE's equal treatment of all data points overlooks differences in regional importance, potentially resulting in high MSE scores that fail to reflect true perceptual resemblance.

In future work, the development of a custom loss function tailored to intricately capture the dynamic complexities present in geomorphic imagery is proposed. Such advancement aims to refine the measure of similarity between modeled and real landscapes, resulting in a more accurate and precise loss function. This enhancement is anticipated to significantly improve calibration accuracy within geomorphological modeling. It is important to highlight that our IMC framework offers flexibility and can readily accommodate alternative evaluation metrics, should they better suit the user's specific requirements.

However, exploring the applicability and effectiveness of the IMC approach in calibrating other physical or numerical models beyond the CL model warrants investigation. Assessing the IMC method's performance across diverse geomorphic environments, spanning various geographical locations and temporal scales, is crucial. Such comprehensive evaluation will illuminate the strengths and potential limitations of the IMC approach when applied to specific geomorphic contexts or environmental settings. Additionally, it would be intriguing to create a synthetic final landscape or DEM. Investigating how the IMC method

autonomously calibrates CL or other numerical geomorphic models to achieve this predetermined end state could offer novel insights into the method's predictive capabilities and its utility in forward modeling geomorphological changes.

*Code and data availability.*   Source code is maintained on GitHub at https://github.com/cbanerji/IMC and the exact version used in this study (including executable code, data and other relevant files ) is archived on Zenodo at https://doi.org/10.5281/zenodo.12747679

**Appendix A**

400 **A1    Parameter list preparation and value selection**

In table A1, we present the exact structure of the PD file for reference. The names of all the parameters that need to be calibrated are included in the top row. In the second row, we include the names of these parameters as represented in the CL configuration XML file, e.g. the parameter "max erode limit" is represented using "maxerodelimit". The next two rows present the numeric upper and lower limits of the IMC search for a certain parameter. Finally, the last two rows present the prior $(\mu, \sigma)$ or (mean,

std) values that define the Gaussian distribution from where the IMC starts its search. The prior (mean) also called the *IMC initial* values can be adjusted with the help of values published in the literature. The prior(std) value is set on intuition and may be updated based on the search space and the scale of values, for a certain parameter.

**Table A1.** Structure and default values of the *Parameter List and Prior data (PD file)*

| Parameter name | Maximum erode limit | In-channel lateral erosion | Vegetation critical shear stress | Min Q | Slope failure threshold | Evaporation rate | Soil creep rate | I/P O/P difference | Slope edge cells | "m" value |
|---|---|---|---|---|---|---|---|---|---|---|
| Parameter name (in CL config. file) | max-erodelimit | lateral-erosionrate | vegcritshear | minq | slopefailure-threshold | evaporation | creeprate | initscans | initialq | mvalue |
| lower-limit | 0.001 | 15 | 80 | 0.001 | 20 | 0.002 | 0.0015 | 0.3 | 0.001 | 0.0057 |
| upper-limit | 0.005 | 25 | 120 | 0.015 | 85 | 0.006 | 0.0035 | 0.7 | 0.1 | 0.02 |
| prior(mean) | 0.003 | 20 | 100 | 0.01 | 50 | 0.004 | 0.0025 | 0.5 | 0.01 | 0.005 |
| prior(std.) | 0.001 | 5 | 10 | 0.001 | 10 | 0.001 | 0.001 | 0.1 | 0.001 | 0.001 |

## A2 Procedure to setup the calibration

### A2.1 Data preparation

The DEMs should be aligned, of the same resolution, and set all no-data values of the DEMs to "-9999". One can use the "setnull" tool from ArcGis-Pro for the same. We tested with DEM rasters that have been converted to Esri ASCII text files (with .txt extension).

### A2.2 Initializing the calibration

As mentioned before the xml file serves as a read-write center for the calibration algorithm and the numerical algorithm, respectively. So we follow the following two-step for initiating the calibration process:

- Prepare your template *xml* and take care of all warnings: Open a CL (orig.) exe and load the template XML file. Next, browse and select each of the relevant DEM and rainfall time series data files. Finally, save the changes back to the XML template and load the data to check for warnings.

  Some parameters also need to be adjusted depending on the data/ DEM and the temporal separation between DEM *year-0* and DEM *year-T*. Calculate hour $(hrs)$ and minute $(mins)$ equivalent of the time difference between the two DEMs. Update parameter "Save file every min." with $mins$, and all other time parameters on "Files" page on CL(orig.). Next on the "Numerical" page, update "max. run duration" with $hrs + 1$. For example, in case, DEM year-0 (= July, 2019) and DEM year-T (= July, 2021), i.e. a difference of3 years, so $hrs = 17544$, $mins = 1052640$.

  Resolve all the warnings and exit.

  These changes can also be made directly in the template XML file through an XML editor but using the CL GUI is more efficient and error-free.

- Use updated template xml: Now this template XML, is updated with the relevant file locations and other data, relevant to the experiment. It should be placed in the Calibration-alg. package and calibration may be initiated from the console.

*Author contributions.* Chayan Banerjee led the research on the topic, designed, and implemented the IMC algorithm, making substantial contributions to the conception and execution of the study. Kien Nguyen, whose vision inspired this work, provided critical insights, played a significant role in drafting the manuscript, and contributed to substantial revisions that enhanced the intellectual content. Clinton Fookes, Gregory Hancock, and Thomas Coulthard significantly contributed through their insightful feedback and thorough review of both the technical content and the overall manuscript. Their rigorous assessments were crucial in ensuring the accuracy, validity, and robustness of the research article.

*Competing interests.* There are no competing interests.

*Acknowledgements.* This research was partly supported by the Advance Queensland Industry Research Fellowship AQIRF024-2021RD4.

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
