# Peer review of "Introducing Iterative Model Calibration (IMC) v1.0: A Generalizable Framework for Numerical Model Calibration with a CAESAR-Lisflood Case Study"

_EGUsphere, 2024_

## Referee Comment (RC1)

Line 12: "retrospective and prospective analyses at various temporal resolutions" is repeated

Line 23: check formatting of citations

Line 42: check formatting of citation, and similar instances throughout the entire manuscript

Table 1: define units of each parameter and in the caption you need to explain the sensitivity scoring that you've provided

Figure 3: label axis using conventional methods e.g. Time (hr), Rainfall (mm), add scale bar to map, and provide legend for DEM values

Section 4.2: describe the DEM data. For example, how was the DEM acquired, what is the spatial resolution and accuracy (vertical and horizontal)

Line 255: need to state earlier in the methods that you will compare uncalibrated, manually calibrated, and automated calibration outputs because introducing this in the results is confusing. Okay, I see that you have introduced this explanation on lines 261-265, but consider providing this before you present the calibration results.

Table 2: erosion volume needs units and state time period of simulation in caption

Fig 5: overall the figure is not clear and contains redundancy. Improvement can be made by visualizing the DODs on top of a hillshade of the 2021 DEM and make topographic changes that are near zero transparent. To provide a fair comparison, make sure that the DOD values have the same range across all maps. Provide units for the legend and a scale bar for the maps. These maps will provide a spatial comparison across calibration methods. The 3D plots are very difficult to understand and should be omitted. Instead a quantitative analysis should be provided by comparing distributions (e.g. density plot) of topographic change per calibration method or another type of visualization (e.g. scatterplots). If you decide to keep the 3D plots, add axis labels and increase the font size of the tick labels. Also, max values on z-axis across all plots should be the same to aid comparison.

Table 3: parameters need units, also check if this table is mentioned in the text.

Line 264: w.r.t.?

Section 4.3.3.: explain in the text what you mean by justifiable landscape changes in the amount of time stated. In addition, state what rainfall are you using for the future, are you just repeating the rainfall file? Overall, I don't understand what you are trying to prove by comparing the output from an uncalibrated and calibrated model. If you are trying to demonstrate that an IMC calibrated model can produce physically plausible predictions you will need observations in the future (e.g. 2023). Without this data your conclusions about the IMC calibrated future output

remain speculative and don't add useful information to the manuscript. Consider rewriting or removing this section.

Figure 6: max value on z-axis across all plots should be the same for a fair comparison. Z-axis needs labels and tick label font size should be larger.

Section 5: what is an ablation study?

Check if figure 7 is referenced in the text.

Figure 8: need labels and units on x-axis

---

## Referee Comment (RC2)

**Comments on manuscript, 'egusphere-2024-1191'**

**General feedback**

First, I have noticed that another reviewer has provided feedback and the authors have updated their manuscript to reflect those comments. **My thoughts below are therefore based on the revised manuscript uploaded on the 6th July 2024**.

The authors present an important and potentially powerful tool to aid in the calibration of complex numerical models. CAESAR-Lisflood, as a model that takes several parameters, many of which can be sensitive to small changes in values, is an ideal model to develop the iterative model calibration (IMC) framework. The conclusion section outlines very effectively how this IMC framework can be adapted for other use cases and in my view, persuasively argues how this is not just confined to CAESAR-Lisflood or modelling gully erosion.

I outline a few core questions/concerns, and list several minor corrections that I recommend the authors should address:

- Why did you choose gully erosion in Australia as your case study to demonstrate your new IMC? I don't doubt there are plenty of good reasons, which may include your familiarity with the environment and/or processes in question, but it wasn't clear to me as a reader why this specific case study is important. Could you elaborate on this in the "Problem statement" section?

- You mention in section 4.1 "Problem statement": simulating at a range of different temporal resolutions (e.g. days, weeks, months – as well as annually). Yet, in terms of observational data, you essentially have just 1 datapoint to compare against – i.e. the net erosion volume and gully morphology differences between the years 2019 & 2021, with no other observations in between. Given the observational data you present, I don't think you can argue that you can accurately simulate changes at fine temporal resolutions. For instance, gully erosion over the 2019-21 time period could be driven by hundreds of small events; a smaller number of intermediate sized events; or 1-2 very large events. You won't know which of these scenarios will apply if you've only got DEMs for 2019 and 2021. I think it would be more appropriate to avoid the subject of modelling at multiple finer temporal resolutions of days, weeks, months, etc.

- You also mention extrapolating beyond 2019-21 and argue that modelled geomorphic changes are "justifiable" and "follow a consistent trend over time". Yet, you don't have the data to back these claims up. I would advise then that you either: a) remove this from the paper entirely, or b) compare your model results to more recent post-2021 observed geomorphic change data – assuming this exists of course!

- You write about "calibration epochs" and describe these as a function of "rounds x iterations". Yet you don't explain clearly what "rounds" and "iterations" mean here. Without this explanation, it's very difficult to interpret the results and discussion you present in section 5.2 "Experimental analysis".

- I don't think you have explained the importance of the experiments to evaluate IMC's efficiency in CL parameter re-estimation. Why does it matter? And are there single optimum values for parameters like "lateral erosion rate"? Perhaps I'm missing something obvious here…

**Specific issues**

Text font on most figures is very small. I have to zoom in quite a lot to see the text on my computer screen and I am certain the text would be unreadable on a printed copy. Please could you make the following amendments?

**Figure 2:** You should move this into section 3. Currently, the figure is introduced at the end of Figure 2, but crucial details like the "prior data (PD) file" get explained in later paragraphs; this current order of presentation is confusing to read.

Also, "CAESAR-Lisflood" is spelt incorrectly on the figure itself.

**Figure 3:** I would consider starting again from scratch with this figure. How about something like this:

Panel (a): A regional map that shows the Bowen Basin of Northern Queensland, with an outline that shows where your specific gully site is located. You could include here an inset map in the top left corner that shows Australia, with a shape or marker point that indicates where Bowen Basin is located. Then panel (b) would show your existing satellite image of the gully site zoomed in – see sketch below that shows what I mean. Below panels (a) and (b), you could have panel (c) showing your pluviometer reading.

[Figure]

You already include the DEMs as part of Figure 4, so I don't think it makes much sense including them in Figure 3. I also think you can give a clearer introduction to the context of your study site to the reader by re-formatting the figure to how I've suggested.

Also, please make sure that the font size of all text on the figure itself is larger than it is currently; it's very difficult to read even when zooming in on a computer screen! You may also want to put the satellite image credit text in the figure caption itself to allow readers to see it properly.

**Figure 4:** Same comments on font size apply here as they do for Figure 3: the numbers on the axes of the graphs depicting erosion volumes need to be enlarged. You may find that this figure needs to be displayed in landscape format to show everything properly.

**Figure 5:** Expand the figure caption slightly to explain what you mean by "standardized standard deviations". You haven't explained what this means anywhere in the main text; adding it to the figure caption would help readers to more quickly interpret this figure at a glance.

**Figure 6:** Please increase the font size of labels on the axes and the vertical colour bar legends.

**Figure 8:** Please increase the font size of labels on axes and the 2 category labels on both plots (i.e. the yellow and orange bars). It may make sense to put part (b) below part (a) to help make the figure more legible overall.

Some parts of the text require further clarification:

In several places in the results, you use the word "Actual" to refer to the observed gully changes. Why not just call it "Observed" instead for clarity?

**Lines 56-59:** You mention that Tsai et al. (2021) propose a couple of data-driven approaches for calibration, yet you only describe one of these. What is the other approach?

**Line 74:** Remove the first word of the sentence "Besides a large number of…"

**Lines 78-79:** I would remove the sentence: "Our calibration approach innovatively leverages…". At this point in the text, you haven't yet demonstrated that this is the case, and it reads more like a sentence for the conclusion section of your paper.

**Lines 92-101:** I don't know if you need this explanation of what each section specifically covers. Perhaps replace with a couple of sentences explaining that you introduce this new IMC algorithm, demonstrate how it works for a chosen landscape evolution modelling context (i.e. gully erosion modelling with C-L), and outline how this IMC could be adapted to other contexts going forward.

**Line 131:** Replace "it's" with "which is".

**Line 155:** Should "Numerical" begin with a capital letter?

**Line 203:** Citation should be written as (Nash and Sutcliffe, 1970).

**Line 219:** You write "…we introduce the study area…", yet you've already introduced the study site in the preceding paragraph. Perhaps delete or move this sentence?

**Line 265:** What does "w.r.t." mean here?

**Line 273:** Surely the "target erosion volume" is derived from the difference between the 2021 DEM and the 2019 DEM, not simply the 2021 DEM itself? Rephrase.

**Table 3 caption:** You mention apparent high variability in parameters 10, 11 & 12. How do you arrive at that judgement? Did you look at the coefficient of variation for all 12 parameters? It might be worth including this as an extra column in your table.

**Line 274:** "We provide a visual comparison of the same in Fig. 4…" Same what exactly?

**Section 4.3.3 Future land evolution: Demonstrative projections:** I don't think you have address Reviewer 1's comments adequately here:

- How have you run future projections here – by repeating the 2019-21 rainfall file?
- How do projected future landscape changes follow a consistent trend over time? Are you suggesting here that the annual rate of erosion does not change compared to the 2019-21 calibration period?
- I don't understand what you mean here by "justifiable landscape changes"? I know you're referring to erosion volumes, but what exactly makes them justifiable here? I agree with

Reviewer 1 that you would likely need to look at more recent observational evidence post-2021 to back up what you're trying to say here.

**Line 291:** What do you mean by "fixed type of numerical model"?

**Line 300:** What do you mean by "rounds" and "iterations"? These terms could easily be used synonymously, so it would be very helpful if you could explain what you mean by each of these.

**Lines 306-309:** I would consider merging with the preceding paragraph. It took me a couple of minutes to work out what you meant by "This phenomenon…"

**Lines 349-351:** You note an important caveat to using MSE as an evaluation metric for your modelling. Is it worth adding a sentence here to mention that your IMC framework could easily be adapted to use an alternative metric instead, should the user feel that that is more appropriate?

---

## Community Comment (CC1)

**Review Response IMC for CL ( Reviewer -2)**

**Reviewer's Comments: -**

First, I have noticed that another reviewer has provided feedback and the authors have updated their manuscript to reflect those comments. My thoughts below are therefore based on the revised manuscript uploaded on the 6th July 2024.

The authors present an important and potentially powerful tool to aid in the calibration of complex numerical models. CAESAR-Lisflood, as a model that takes several parameters, many of which can be sensitive to small changes in values, is an ideal model to develop the iterative model calibration (IMC) framework. The conclusion section outlines very effectively how this IMC framework can be adapted for other use cases and in my view, persuasively argues how this is not just confined to CAESAR-Lisflood or modelling gully erosion.

***- Response: We have addressed all the reviewer's concerns and appreciate the valuable feedback. All major changes made to the manuscript in response to the 2nd reviewer's comments are in purple font.***

- I outline a few core questions/concerns, and list several minor corrections that I recommend the authors should address:

- Why did you choose gully erosion in Australia as your case study to demonstrate your new IMC? I don't doubt there are plenty of good reasons, which may include your familiarity with the environment and/or processes in question, but it wasn't clear to me as a reader why this specific case study is important. Could you elaborate on this in the "Problem statement" section?

  ***- Response:*** *Thank you for the comment. We have now provided a justification for selecting gully erosion in Australia as the focus of our study*

- You mention in section 4.1 "Problem statement": simulating at a range of different temporal resolutions (e.g. days, weeks, months – as well as annually). Yet, in terms of observational data, you essentially have just 1 datapoint to compare against – i.e. the net erosion volume and gully morphology differences between the years 2019 & 2021, with no other observations in between. Given the observational data you present, I don't think you can argue that you can accurately simulate changes at fine temporal resolutions. For instance, gully erosion over the 2019-21 time period could be driven by hundreds of small events; a smaller number of intermediate sized events; or 1-2 very large events. You won't know which of these scenarios will apply if you've only got DEMs for 2019 and 2021. I think it would be more appropriate to avoid the subject of modelling at multiple finer temporal resolutions of days, weeks, months, etc.

  ***- Response:*** *Thank you for the comment. We have now removed all related arguments and statements from the manuscript.*

- You also mention extrapolating beyond 2019-21 and argue that modelled geomorphic changes are "justifiable" and "follow a consistent trend over time". Yet, you don't have the data to back these claims up. I would advise then that you either: a) remove this from the paper entirely, or b) compare your model results to more recent post-2021 observed geomorphic change data –assuming this exists of course!

  *- **Response:** Thank you for the comment. We have now removed all related arguments and statements from the manuscript.*

- You write about "calibration epochs" and describe these as a function of "rounds x iterations". Yet you don't explain clearly what "rounds" and "iterations" mean here. Without this explanation, it's very difficult to interpret the results and discussion you present in section 5.2 "Experimental analysis".

  *- **Response:** Thank you for the comment. We have now included detailed explanation of the terms 'round' and 'iteration' in Section 3.1. To avoid confusion, we have removed 'calibration epoch' with 'calibration run'.*

- I don't think you have explained the importance of the experiments to evaluate IMC's efficiency in CL parameter re-estimation. Why does it matter? And are there single optimum values for parameters like "lateral erosion rate"? Perhaps I'm missing something obvious here…

  *- **Response:** Thank you for the comment. We have now included a detailed explanation on the importance of the experiments in the beginning of Section-5.3. To avoid confusion, we have replaced the term 'optimal value' with 'benchmark', which is the target value being re-estimated using IMC.*

- **Specific issues**
  Text font on most figures is very small. I have to zoom in quite a lot to see the text on my computer screen and I am certain the text would be unreadable on a printed copy. Please could you make the following amendments?

- Figure 2: You should move this into section 3. Currently, the figure is introduced at the end of Figure 2, but crucial details like the "prior data (PD) file" get explained in later paragraphs; this current order of presentation is confusing to read. Also, "CAESAR-Lisflood" is spelt incorrectly on the figure itself.

  *- **Response:** Thank you for the comment. We have now moved the Figure-2 to inside Section:3 and corrected the spelling error in the image. We have also increased the font size of all labels and texts on figures.*

- Figure 3: I would consider starting again from scratch with this figure. How about something like this:

  Panel (a): A regional map that shows the Bowen Basin of Northern Queensland, with an outline that shows where your specific gully site is located. You could include here an inset map in the top left corner that shows Australia, with a shape or marker point that indicates where Bowen Basin is located. Then panel (b) would show your existing satellite image of the gully site zoomed in – see sketch below that shows what I mean. Below panels (a) and (b), you could have panel (c) showing your pluviometer reading.

[Figure]

[Figure]

  You already include the DEMs as part of Figure 4, so I don't think it makes much sense including them in Figure 3. I also think you can give a clearer introduction to the context of your study site to the reader by re-formatting the figure to how I've suggested.

  Also, please make sure that the font size of all text on the figure itself is larger than it is currently; it's very difficult to read even when zooming in on a computer screen! You may also want to put the satellite image credit text in the figure caption itself to allow readers to see it properly.

  *- **Response:** Thank you for your comment and detailed restructuring of the Figure. We have now completely modified the figure based on your suggestions. We have increased the font sizes of all text labels and moved the image credits to the caption.*

- Figure 4: Same comments on font size apply here as they do for Figure 3: the numbers on the axes of the graphs depicting erosion volumes need to be enlarged. You may find that this figure needs to be displayed in landscape format to show everything properly.

  *- **Response:** Thank you for the comment. We have reworked on the figure, increasing the font sizes of legends and labels, such that they are readable on paper. We have done some restructuring of the legends and colour-bars to make individual figures larger and better visible. We found that a portrait configuration serves a better fit for this modified multi-panel figure.*

- Figure 5: Expand the figure caption slightly to explain what you mean by "standardized standard deviations". You haven't explained what this means anywhere in the main text; adding it to the figure caption would help readers to more quickly interpret this figure at a glance.

  *- **Response:** Thank you for the comment. We have now extended the caption of Fig. 5 to include a brief introduction of the metric Standardized Standard deviation (SSDev), for quick interpretation.*

- Figure 6: Please increase the font size of labels on the axes and the vertical colour bar legends.

  *- **Response:** Thank you for the comment. Considering comments from both the reviewers we have removed the 'Section 4.3.3 Future land evolution', and the concerning figure (Fig.6 in previous draft).*

- Figure 8: Please increase the font size of labels on axes and the 2 category labels on both plots (i.e. the yellow and orange bars). It may make sense to put part (b) below part (a) to help make the figure more legible overall.

  *- **Response:** Thank you for the comment. We have made the advised changes in the figure (Fig. 7 in present draft).*

  **Some parts of the text require further clarification:**

- In several places in the results, you use the word "Actual" to refer to the observed gully changes. Why not just call it "Observed" instead for clarity?

  *- **Response:** Thank you for the comment. We have now replaced the word 'Actual' with 'Observed' for better clarity.*

- Lines 56-59: You mention that Tsai et al. (2021) propose a couple of data-driven approaches for calibration, yet you only describe one of these. What is the other approach?

  *- **Response:** Thank you for the comment. We have now updated the concerned sentences.*

- Line 74: Remove the first word of the sentence "Besides a large number of..."

  *- **Response:** Thank you for the comment. The mentioned issue was corrected.*

- Lines 78-79: I would remove the sentence: "Our calibration approach innovatively leverages...". At this point in the text, you haven't yet demonstrated that this is the case, and it reads more like a sentence for the conclusion section of your paper.

  *- **Response:** Thank you for the comment. We have now rephrased the sentence saying 'Our calibration approach aims to leverage limited DEM data ....'*

- Lines 92-101: I don't know if you need this explanation of what each section specifically covers. Perhaps replace with a couple of sentences explaining that you introduce this new IMC algorithm, demonstrate how it works for a chosen landscape evolution modelling context (i.e. gully erosion modelling with C-L), and outline how this IMC could be adapted to other contexts going forward.

  *- Response: Thank you for your valuable feedback. We believe that including an introduction to specific sections enhances the reader's experience by improving clarity, comprehension, and engagement, while also facilitating easier navigation and providing a clear framework for the content that follows.*

  *However, we have also incorporated your advice by including a summary before the concerned section as a summary insight into the upcoming sections.*

- Line 131: Replace "it's" with "which is".

  *- Response: Thank you for the comment. The mentioned issue was corrected.*

- Line 155: Should "Numerical" begin with a capital letter?

  *- Response: Thank you for the comment. The mentioned issue was corrected.*

- Line 203: Citation should be written as (Nash and Sutcliffe, 1970).

  *- Response: Thank you for the comment. The mentioned issue was corrected.*

- Line 219: You write "...we introduce the study area...", yet you've already introduced the study site in the preceding paragraph. Perhaps delete or move this sentence?

  *- Response: Thank you for the comment. We have now moved the study area figures after the mentioned statement in Section-4.3.1*

- Line 265: What does "w.r.t." mean here?

  *- Response: Thank you for the comment. The mentioned word is now replaced to avoid confusion.*

- Line 273: Surely the "target erosion volume" is derived from the difference between the 2021 DEM and the 2019 DEM, not simply the 2021 DEM itself? Rephrase

  *- Response: Thank you for the comment. We have now corrected this error.*

- Table 3 caption: You mention apparent high variability in parameters 10, 11 & 12. How do you arrive at that judgement? Did you look at the coefficient of variation for all 12 parameters? It might be worth including this as an extra column in your table.

- *Response: Thank you for the comment. We have now introduced a 'coefficient of variation' column in the table and included a brief introduction of this metric in the caption.*

- Line 274: "We provide a visual comparison of the same in Fig. 4..." Same what exactly?
  - *Response: Thank you for the comment. We have revised the section in question to enhance clarity, which is now reflected at the end of Section 4. Here we mentioned that 'In Fig.4, we further elaborated on the numerical results presented in Table2 through extensive visual comparison'.*

- Section 4.3.3 Future land evolution: Demonstrative projections: I don't think you have address Reviewer 1's comments adequately here:
  How have you run future projections here – by repeating the 2019-21 rainfall file?
  How do projected future landscape changes follow a consistent trend over time? Are you suggesting here that the annual rate of erosion does not change compared to the 2019-21 calibration period?
  I don't understand what you mean here by "justifiable landscape changes"? I know you're referring to erosion volumes, but what exactly makes them justifiable here? I agree with Reviewer 1 that you would likely need to look at more recent observational evidence post 2021 to back up what you're trying to say here.
  - *Response: Thank you for the comment. Considering your advice and that of the previous reviewer we have withdrawn the whole section.*

- Line 291: What do you mean by "fixed type of numerical model"?
  - *Response: Thank you for the comment. The mentioned word is now with 'a particular type. . .' to avoid confusion.*

- Line 300: What do you mean by "rounds" and "iterations"? These terms could easily be used synonymously, so it would be very helpful if you could explain what you mean by each of these.
  - *Response: Thank you for the comment. The mentioned words are now explained in details in Section 3.1., for better clarity.*

- Lines 306-309: I would consider merging with the preceding paragraph. It took me a couple of minutes to work out what you meant by "This phenomenon..."
  - *Response: Thank you for the comment. We have now merged the paragraphs as advised.*

- Lines 349-351: You note an important caveat to using MSE as an evaluation metric for your modelling. Is it worth adding a sentence here to mention that your IMC framework

could easily be adapted to use an alternative metric instead, should the user feel that that is more appropriate?

*- **Response:** Thank you for the comment. We have now included a sentence mentioning 
[revised manuscript text omitted]

---

## Author Comment (AC2)

**Review Response IMC for CL**

All the modified sections are kept in blue font.

***Reviewer's Comments:***

1. ***Line 12: "retrospective and prospective analyses at various temporal resolutions" is repeated***
   Thank you for your comment. The error is now corrected.

2. ***Line 23: check formatting of citations***
   Thank you for your comment. The citations are now properly formatted based on the guidelines from the Journal's submission page.

3. ***Line 42: check formatting of citation, and similar instances throughout the entire manuscript***
   Thank you for your comment. The citations are now properly formatted based on the guidelines from the Journal's submission page.

4. ***Table 1: define units of each parameter and in the caption, you need to explain the sensitivity scoring that you've provided***
   Thank you for your comment.
   The caption of Table1, is updated with explanation of the sensitivity scoring. The table entries are also updated with units of the parameters wherever possible since many of them are dimensionless.

5. ***Figure 3: label axis using conventional methods e.g. Time (hr), Rainfall (mm), add scale bar to map, and provide legend for DEM values***
   Thank you for your comment.
   We have now updated the rainfall data with standard x and y labels, added scale bar to the map and included legend for the DEM.

6. ***Section 4.2: describe the DEM data. For example, how was the DEM acquired, what is the spatial resolution and accuracy (vertical and horizontal)***
   Thank you for your comment.
   All relevant information related to the acquisition of the DEMs, resolution and accuracy is now included in section:4.2

7. ***Line 255: need to state earlier in the methods that you will compare uncalibrated, manually calibrated, and automated calibration outputs because introducing this in the results is confusing. Okay, I see that you have introduced this explanation on lines 261-265, but consider providing this before you present the calibration results.***
   Thank you for your comment.
   We have now included the mentioned information before the presentation of calibration results for better understanding.

8. ***Table 2: erosion volume needs units and state time period of simulation in caption***
   Thank you for your comment.
   The table is updated with the unit for erosion volume. The time-period details of the calibration epochs are now included in the caption.

9. ***Fig 5: overall the figure is not clear and contains redundancy. Improvement can be made by visualizing the DODs on top of a hill-shade of the 2021 DEM and make topographic changes that are near zero transparent. To provide a fair comparison, make sure that the DOD values have the same range across all maps. Provide units for the legend and a scale bar for the maps. These maps will provide a spatial comparison across calibration methods.***
   We have now replaced the 2D DOD column in the figure with the advised visualization of placing DODs on hill-shade of 2021 DEM and setting the near zero volume to transparent. The legends and symbology value ranges are kept same and scales provided in each map.

10. ***The 3D plots are very difficult to understand and should be omitted. Instead, a quantitative analysis should be provided by comparing distributions (e.g. density plot) of topographic change per calibration method or another type of visualization (e.g. scatterplots). If you decide to keep the 3D plots, add axis labels and increase the font size of the tick labels. Also, max values on z-axis across all plots should be the same to aid comparison.***
    Thank you for your comment.
    We have retained the 3D plots and have modified them addressing your concerns. We have added the axis label, increased the font size and made the z-axis max value same across all the plots for easy comparison.

11. ***Table 3: parameters need units, also check if this table is mentioned in the text.***
    Thank you for your comment. The parameters were updated with units and the table is now referenced in the text.

**12. Line 264: w.r.t.?**

13. This error is now corrected.

**14. Section 4.3.3.: explain in the text what you mean by justifiable landscape changes in the amount of time stated. In addition, state what rainfall are you using for the future, are you just repeating the rainfall file? Overall, I don't understand what you are trying to prove by comparing the output from an uncalibrated and calibrated model. If you are trying to demonstrate that an IMC calibrated model can produce physically plausible predictions, you will need observations in the future (e.g. 2023). Without this data your conclusions about the IMC calibrated future output remain speculative and don't add useful information to the manuscript. Consider rewriting or removing this section.**

Thank you for your comment.

We have now rewritten and retitled the subsection (4.3.3.). The images are demonstrative projections and illustrates that IMC calibrated CL can generate consistent and logical predictions of present and future land evolution compared to uncalibrated approaches.

The predicted landscape changes follow consistent trend over time (compare with current predictions i.e. year 2021 in Fig 4p), aligning with the natural progression of erosion.

**15. Figure 6: max value on z-axis across all plots should be the same for a fair comparison. Z-axis needs labels and tick label font size should be larger.**

Thank you for your comment.

All the plots in Fig.6, now have been updated with all having the same maximum z-axis value. Separate Colorbars have also been added for each plot.

**16. Section 5: what is an ablation study?**

'Ablation study' is a term popularly used in Machine Learning and Computer science disciplines. It is a research method used to evaluate the impact of different components or features of a system, model, or process by systematically removing or altering them and observing the effects on performance.

We have now replaced it with 'Experimental study' to avoid any confusion.

**17. Check if figure 7 is referenced in the text.**

Thank you for pointing out the error.

We have now corrected it and included the reference to Fig.7 in text.

**18. Figure 8: need labels and units on x-axis**

Thank you for pointing out the error.

We have now updated the figures with relevant x-axis label and unit.

[revised manuscript text omitted]
 across unseen temporal resolutions using CL. The objective is to illustrate that the IMC-calibrated CL can generate consistent and logical predictions of present and future land evolution compared to uncalibrated approaches. The predicted landscape changes follow a consistent trend over time (compare with current predictions i.e. year 2021 in Fig 4p), aligning with the natural progression of erosion. However, we do not test the accuracy of these predictions here. In this demonstration, the CL parameters are 
[revised manuscript text omitted]

---

## Author Response (AR1)

**Review Response IMC for CL (Reviewer -2) **[ round I ]**

**Reviewer's Comments: -**
First, I have noticed that another reviewer has provided feedback and the authors have updated their manuscript to reflect those comments. My thoughts below are therefore based on the revised manuscript uploaded on the 6th July 2024.

The authors present an important and potentially powerful tool to aid in the calibration of complex numerical models. CAESAR-Lisflood, as a model that takes several parameters, many of which can be sensitive to small changes in values, is an ideal model to develop the iterative model calibration (IMC) framework. The conclusion section outlines very effectively how this IMC framework can be adapted for other use cases and in my view, persuasively argues how this is not just confined to CAESAR-Lisflood or modelling gully erosion.
*- Response: We have addressed all the reviewer's concerns and appreciate the valuable feedback. All major changes made to the manuscript in response to the 2$^{nd}$ reviewer's comments are in purple font.*

- I outline a few core questions/concerns, and list several minor corrections that I recommend the authors should address:
- Why did you choose gully erosion in Australia as your case study to demonstrate your new IMC? I don't doubt there are plenty of good reasons, which may include your familiarity with the environment and/or processes in question, but it wasn't clear to me as a reader why this specific case study is important. Could you elaborate on this in the "Problem statement" section?
*- Response: Thank you for the comment. We have now provided a justification for selecting gully erosion in Australia as the focus of our study*

- You mention in section 4.1 "Problem statement": simulating at a range of different temporal resolutions (e.g. days, weeks, months – as well as annually). Yet, in terms of observational data, you essentially have just 1 datapoint to compare against – i.e. the net erosion volume and gully morphology differences between the years 2019 & 2021, with no other observations in between. Given the observational data you present, I don't think you can argue that you can accurately simulate changes at fine temporal resolutions. For instance, gully erosion over the 2019-21 time period could be driven by hundreds of small events; a smaller number of intermediate sized events; or 1-2 very large events. You won't know which of these scenarios will apply if you've only got DEMs for 2019 and 2021. I think it would be more appropriate to avoid the subject of modelling at multiple finer temporal resolutions of days, weeks, months, etc.
*- Response: Thank you for the comment. We have now removed all related arguments and statements from the manuscript.*

- You also mention extrapolating beyond 2019-21 and argue that modelled geomorphic changes are "justifiable" and "follow a consistent trend over time". Yet, you don't have the data to back these claims up. I would advise then that you either: a) remove this from the paper entirely, or b) compare your model results to more recent post-2021 observed geomorphic change data –assuming this exists of course!

  *- Response: Thank you for the comment. We have now removed all related arguments and statements from the manuscript.*

- You write about "calibration epochs" and describe these as a function of "rounds x iterations". Yet you don't explain clearly what "rounds" and "iterations" mean here. Without this explanation, it's very difficult to interpret the results and discussion you present in section 5.2 "Experimental analysis".

  *- Response: Thank you for the comment. We have now included detailed explanation of the terms 'round' and 'iteration' in Section 3.1. To avoid confusion, we have removed 'calibration epoch' with 'calibration run'.*

- I don't think you have explained the importance of the experiments to evaluate IMC's efficiency in CL parameter re-estimation. Why does it matter? And are there single optimum values for parameters like "lateral erosion rate"? Perhaps I'm missing something obvious here...

  *- Response: Thank you for the comment. We have now included a detailed explanation on the importance of the experiments in the beginning of Section-5.3. To avoid confusion, we have replaced the term 'optimal value' with 'benchmark', which is the target value being re-estimated using IMC.*

- **Specific issues**
  Text font on most figures is very small. I have to zoom in quite a lot to see the text on my computer screen and I am certain the text would be unreadable on a printed copy. Please could you make the following amendments?
- Figure 2: You should move this into section 3. Currently, the figure is introduced at the end of Figure 2, but crucial details like the "prior data (PD) file" get explained in later paragraphs; this current order of presentation is confusing to read. Also, "CAESAR-Lisflood" is spelt incorrectly on the figure itself.

  *- Response: Thank you for the comment. We have now moved the Figure-2 to inside Section:3 and corrected the spelling error in the image. We have also increased the font size of all labels and texts on figures.*

- Figure 3: I would consider starting again from scratch with this figure. How about something like this:
  Panel (a): A regional map that shows the Bowen Basin of Northern Queensland, with an outline that shows where your specific gully site is located. You could include here an inset map in the top left corner that shows Australia, with a shape or marker point that indicates where Bowen Basin is located. Then panel (b) would show your existing satellite image of the gully site zoomed in – see sketch below that shows what I mean. Below panels (a) and (b), you could have panel (c) showing your pluviometer reading.

[Figure]

[Figure]

  You already include the DEMs as part of Figure 4, so I don't think it makes much sense including them in Figure 3. I also think you can give a clearer introduction to the context of your study site to the reader by re-formatting the figure to how I've suggested.
  Also, please make sure that the font size of all text on the figure itself is larger than it is currently; it's very difficult to read even when zooming in on a computer screen! You may also want to put the satellite image credit text in the figure caption itself to allow readers to see it properly.
  ***- Response:*** *Thank you for your comment and detailed restructuring of the Figure. We have now completely modified the figure based on your suggestions. We have increased the font sizes of all text labels and moved the image credits to the caption.*

- Figure 4: Same comments on font size apply here as they do for Figure 3: the numbers on the axes of the graphs depicting erosion volumes need to be enlarged. You may find that this figure needs to be displayed in landscape format to show everything properly.
  ***- Response:*** *Thank you for the comment. We have reworked on the figure, increasing the font sizes of legends and labels, such that they are readable on paper. We have done some restructuring of the legends and colour-bars to make individual figures larger and better visible. We found that a portrait configuration serves a better fit for this modified multi-panel figure.*

- Figure 5: Expand the figure caption slightly to explain what you mean by "standardized standard deviations". You haven't explained what this means anywhere in the main text; adding it to the figure caption would help readers to more quickly interpret this figure at a glance.

  *- **Response:** Thank you for the comment. We have now extended the caption of Fig. 5 to include a brief introduction of the metric Standardized Standard deviation (SSDev), for quick interpretation.*

- Figure 6: Please increase the font size of labels on the axes and the vertical colour bar legends.

  *- **Response:** Thank you for the comment. Considering comments from both the reviewers we have removed the 'Section 4.3.3 Future land evolution', and the concerning figure (Fig.6 in previous draft).*

- Figure 8: Please increase the font size of labels on axes and the 2 category labels on both plots (i.e. the yellow and orange bars). It may make sense to put part (b) below part (a) to help make the figure more legible overall.

  *- **Response:** Thank you for the comment. We have made the advised changes in the figure (Fig. 7 in present draft).*

  **Some parts of the text require further clarification:**

- In several places in the results, you use the word "Actual" to refer to the observed gully changes. Why not just call it "Observed" instead for clarity?

  *- **Response:** Thank you for the comment. We have now replaced the word 'Actual' with 'Observed' for better clarity.*

- Lines 56-59: You mention that Tsai et al. (2021) propose a couple of data-driven approaches for calibration, yet you only describe one of these. What is the other approach?

  *- **Response:** Thank you for the comment. We have now updated the concerned sentences.*

- Line 74: Remove the first word of the sentence "Besides a large number of…"

  *- **Response:** Thank you for the comment. The mentioned issue was corrected.*

- Lines 78-79: I would remove the sentence: "Our calibration approach innovatively leverages…". At this point in the text, you haven't yet demonstrated that this is the case, and it reads more like a sentence for the conclusion section of your paper.

  *- **Response:** Thank you for the comment. We have now rephrased the sentence saying 'Our calibration approach aims to leverage limited DEM data ….'*

- Lines 92-101: I don't know if you need this explanation of what each section specifically covers. Perhaps replace with a couple of sentences explaining that you introduce this new IMC algorithm, demonstrate how it works for a chosen landscape evolution modelling context (i.e. gully erosion modelling with C-L), and outline how this IMC could be adapted to other contexts going forward.
  - *- Response: Thank you for your valuable feedback. We believe that including an introduction to specific sections enhances the reader's experience by improving clarity, comprehension, and engagement, while also facilitating easier navigation and providing a clear framework for the content that follows.*
  *However, we have also incorporated your advice by including a summary before the concerned section as a summary insight into the upcoming sections.*

- Line 131: Replace "it's" with "which is".
  - *- Response: Thank you for the comment. The mentioned issue was corrected.*

- Line 155: Should "Numerical" begin with a capital letter?
  - *- Response: Thank you for the comment. The mentioned issue was corrected.*

- Line 203: Citation should be written as (Nash and Sutcliffe, 1970).
  - *- Response: Thank you for the comment. The mentioned issue was corrected.*

- Line 219: You write "…we introduce the study area…", yet you've already introduced the study site in the preceding paragraph. Perhaps delete or move this sentence?
  - *- Response: Thank you for the comment. We have now moved the study area figures after the mentioned statement in Section-4.3.1*

- Line 265: What does "w.r.t." mean here?
  - *- Response: Thank you for the comment. The mentioned word is now replaced to avoid confusion.*

- Line 273: Surely the "target erosion volume" is derived from the difference between the 2021 DEM and the 2019 DEM, not simply the 2021 DEM itself? Rephrase
  - *- Response: Thank you for the comment. We have now corrected this error.*

- Table 3 caption: You mention apparent high variability in parameters 10, 11 & 12. How do you arrive at that judgement? Did you look at the coefficient of variation for all 12 parameters? It might be worth including this as an extra column in your table.

- *Response: Thank you for the comment. We have now introduced a 'coefficient of variation' column in the table and included a brief introduction of this metric in the caption.*

- Line 274: "We provide a visual comparison of the same in Fig. 4..." Same what exactly?
  - *Response: Thank you for the comment. We have revised the section in question to enhance clarity, which is now reflected at the end of Section 4. Here we mentioned that 'In Fig.4, we further elaborated on the numerical results presented in Table2 through extensive visual comparison'.*

- Section 4.3.3 Future land evolution: Demonstrative projections: I don't think you have address Reviewer 1's comments adequately here:
  How have you run future projections here – by repeating the 2019-21 rainfall file?
  How do projected future landscape changes follow a consistent trend over time? Are you suggesting here that the annual rate of erosion does not change compared to the 2019-21 calibration period?
  I don't understand what you mean here by "justifiable landscape changes"? I know you're referring to erosion volumes, but what exactly makes them justifiable here? I agree with Reviewer 1 that you would likely need to look at more recent observational evidence post 2021 to back up what you're trying to say here.
  - *Response: Thank you for the comment. Considering your advice and that of the previous reviewer we have withdrawn the whole section.*

- Line 291: What do you mean by "fixed type of numerical model"?
  - *Response: Thank you for the comment. The mentioned word is now with 'a particular type. . .' to avoid confusion.*

- Line 300: What do you mean by "rounds" and "iterations"? These terms could easily be used synonymously, so it would be very helpful if you could explain what you mean by each of these.
  - *Response: Thank you for the comment. The mentioned words are now explained in detail in Section 3.1., for better clarity.*

- Lines 306-309: I would consider merging with the preceding paragraph. It took me a couple of minutes to work out what you meant by "This phenomenon..."
  - *Response: Thank you for the comment. We have now merged the paragraphs as advised.*

- Lines 349-351: You note an important caveat to using MSE as an evaluation metric for your modelling. Is it worth adding a sentence here to mention that your IMC framework

could easily be adapted to use an alternative metric instead, should the user feel that that is more appropriate?

*- **Response:** Thank you for the comment. We have now included a sentence mentioning that '... our IMC framework offers flexibility and can readily accommodate alternative evaluation metrics, should they better suit the user's specific requirements.'*

**Review Response IMC for CL (Reviewer -2) **[ round II]**

**Reviewer's Comments: -**

I would like to thank the authors for sharing the latest iteration of their manuscript. I am pleased to see that most of my suggestions have been addressed, and in my opinion, this paper is in a much stronger position to be accepted for full publication.

*- Response: Thank you for your comments and useful insights. The changes are in 'cyan' colour.*

- A small comment from me: You still write in the 1st paragraph of the problem statement (section 4.1) that the "...secondary objective or application of this calibrated numerical model is to predict the landscape evolution of past or future years (or days or months)..." and in the 2nd paragraph of section 4.1: "stSecond, we want the calibrated-CL to perform interpolation and generate DEM data for different temporal resolutions like days, weeks, months, and years, within or outside (past/future) the 2019-2021 period". But you've said in your reply to my comments that these statements had been removed. Could you remove these please?

  Perhaps you should re-write section 4.1 to explain briefly what you intend to do for section 5 - i.e. comparing calibration approaches; experiments with different lengths of calibration runs; evaluating IMC's Efficiency in CL parameter re-estimation?

  *- Response:* We have now rewritten the mentioned Section: 4.1. (see lines 222-230). Additionally, we have made a minor change to the abstract (see lines 10-12).

[revised manuscript text omitted]

---

## Author Response (AR2)

**Report #2** (Review Response)

**Reviewer's Comments:** *I've previously reviewed this manuscript in the preprint stage and have no major issues with this version.*

**Response:** Thank you for your continued engagement with our manuscript. We appreciate your feedback in improving the paper.

- **Reviewer's Comments:** *My only suggestion is to redo figure 4 as it is very redundant, especially the presentation of each DEM, which each look the same. Perhaps you can include the observed DEMs in figure 3 and only keep either the DoD maps or 3d plots in figure 4.*

**Response:** Thank you for your suggestion. We have now updated both the Figures 3 and 4. We have included the observed DEMs in Fig.3 and removed it from Fig.4.

We have decided to retain both the DoD maps and 3-D plots in the Fig.4, since each visualization technique contributes unique insights and enhances the interpretative power of the data, thereby offering a more complete picture of landscape dynamics than either method could provide alone.

- **Reviewer's Comments:** *I'd also be interested in seeing scatterplots of the observed and simulated elevation values or the distributions of DoD values and some discussion on this.*

**Response:** Thank you for your suggestion.

We have now added a plot comparing the distribution of DoD values from all calibration approaches with the observed DoD. We have also included a discussion (see lines 298 to 312) to explain this comparison. We chose a boxplot for its concise visual summary of central tendencies, spread, and outlier behaviour across calibration settings, enabling us to assess alignment with the observed DoD and identify systematic biases or deviations.

To effectively summarize statistical variations in DoD, we use boxplots representing 1D vectors derived from flattened 2D DoDs. We present both the scatter and the boxplot below.

[Figure]

- **Reviewer's Comments:** *Lastly, in figure 4 why are you only plotting erosion values, don't you have deposition too somewhere on the landscape?*

**Response:** We focus on plotting erosion values since erosion processes often dominate landscape evolution, leading to clearer and more measurable outcomes. This approach allows us to establish precise calibration targets and identify critical areas at risk of degradation. While deposition is indeed present in the landscape, it can be more diffused and challenging to quantify and be used in the calibration process. By emphasizing erosion, we enhance model performance and address data limitations, as there are typically more available observational data on erosion.